# Analytical Delay Modeling for a Sub-Threshold Cell Circuit with the Inverse Gaussian Distribution Function

**Jiuyue Wang** [1,2], **Yuping Wu** [1,2,3,4,*] **, Xuelian Zhang** [2,3] **, Zhiqiang Li** [2] **and Lan Chen** [2,3]

1   University of Chinese Academy of Sciences, Beijing 100049, China
2   Institute of Microelectronics, Chinese Academy of Sciences, Beijing 100029, China
3   Beijing Key Laboratory of Three Dimensional and Nanometer Integrated Circuit Design Automation Technology, Beijing 100029, China
4   Key Laboratory of Microelectronic Devices & Integrated Technology, Chinese Academy of Sciences, Beijing 100029, China
*   Correspondence: wuyuping@ime.ac.cn; Tel.: +86-010-8299-5693

**Abstract:** Considering that power consumption (PC) is an extremely important indicator in digital circuit design, lower PC has always been our pursuit. PC and power supply voltage are positively correlated, and in this case, we must reduce the operating voltage of the circuit. However, as the voltage continues to decrease, various secondary effects and process variations become increasingly influential, making the delay distribution and its statistical characteristics more difficult to predict. In this paper, an inverse Gaussian distribution is used to model the propagation delay. Taking into account the local process variation, the multi-input delay analytical expression is derived according to the sub-threshold current formula to accurately predict the distribution and statistical characteristics of the delay, and the delay is obtained by calculation instead of Monte Carlo simulation, which greatly reduces the simulation time. The accuracy of the delay expression and delay distribution have been tested under 22 nm FDSOI technology and good results were obtained with operating voltages from 0.20 V to 0.30 V, in which the mean error of the delay is approx. 1.5%, the variance error is approx. 4.3%, and the error of the cumulative distribution function is approx. 2%.

**Keywords:** delay distribution; statistical characteristics; inverse Gaussian distribution; sub-threshold

## 1. Introduction

With transistor sizes continuing to shrink and the need to reduce power consumption, complementary metal–oxide semiconductor (CMOS) circuits under sub-threshold logic are becoming increasingly used [1,2]. Since power consumption and operating voltage are positively correlated, the sub-threshold region provides better energy efficiency compared to the near-threshold region and super-threshold region [3]. However, as the operating voltage decreases, various secondary effects and process variations become increasingly influential. The primary sources of process variations that affect device performance are random dopant fluctuations (RDFs) and channel length variation [3], in which the RDFs mainly cause the variations in the threshold voltage for transistors, and the channel length variation affects the electrical properties, increasing the threshold voltage for short channel devices. Moreover, there are many other variations, including mobility fluctuation, channel width variation, oxide charge variation, and so on. In the sub-threshold region, the propagation delay is much larger than that in other regions due to process variations [4], and its prediction faces great challenges. Under normal voltages, the delay of the cell circuits follows the Gaussian distribution, as shown in Figure 1a. However, the delay has a nonlinear relationship with the variation in the sub-threshold region, resulting in the delay distribution being difficult to predict, as shown in Figure 1b.

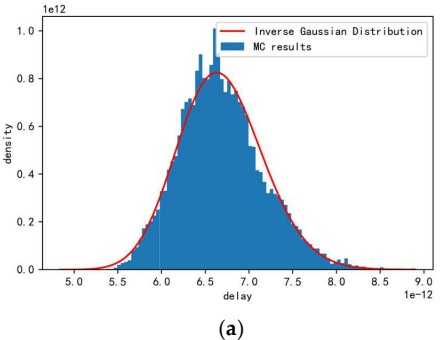 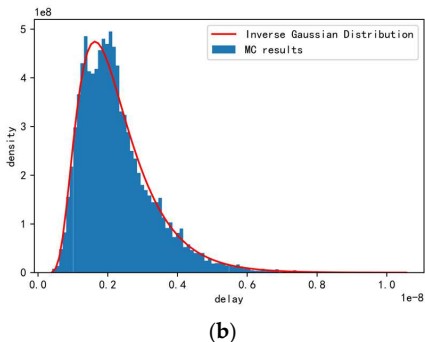

(**a**)          (**b**)

**Figure 1.** PDF: (**a**) 0.6 V (super-threshold); (**b**) 0.2 V (sub-threshold).

At present, the authoritative method is to use Monte Carlo (MC) SPICE simulation to obtain the propagation delay; however, this method requires a lot of time in each state, so it is necessary to find a suitable method to accurately and quickly predict the gate delay and its distribution. For this, some scholars have undertaken extensive research in this field. The paper [5] presented an error-aware model for arithmetic and logic circuits that accurately estimated the propagation delay of output bits in a digital module, but its operation was time-consuming. In [3], Abu-Rahma studied the effect of random fluctuation variation on the gate delay variation and derived a simple and scalable statistical model to efficiently estimate delay variation at conventional and ultra-threshold voltages. Caltech derived a delay model suitable for the near-threshold region (NTV), and obtained good results [6]. In [7], the authors derived a new simplified drain current model to clarify the relationship between supply voltage (near threshold) and delay, and analyzed its statistical characteristics with a logarithmic distribution; however, the error was unacceptable (13%). In [8], a statistical timing model for a CMOS inverter was proposed in the NTV region under process variation considering fast and slow input, which was derived analytically with a novel segmented step approximation method to overcome the integral issue of the drain current equation for ramp input. In 2019, Southeast University established a statistical model of near-threshold drain current and gate delay based on a logarithmic skew normal (LSN) distribution by using moment matching technology, and the prediction sensitivity error of gate delay was less than 8% [9]. The studies [10–12] used machine learning methods to model the delay and obtained good results; nevertheless, this still required a large data set, and it did not physically explain the relationship between the delay and various parameters.

This paper first proposes a probability density function suitable for the delay distribution in the sub-threshold region, i.e., the inverse Gaussian distribution, and indicates the required modeling parameters, see Section 2. Then, according to the classic sub-threshold current expression, the transition time variable is introduced, which is divided into two parts: fast input and slow input, and the equation is constructed by Kirchhoff's law. Following this, the analytical expression of the mean delay and variance in the case of multiple inputs is derived, as detailed in Section 3. Section 4 verifies the derived expression and predicts the delay distribution and related error calculations with the calculation results. Section 5 summarizes all the study.

## 2. Statistical Distribution Model of Delay

The inverse Gaussian distribution (IGD) is a commonly used distribution in statistics [13] with its density function given in Equation (1):

$$f(x, \mu, \lambda) = \left[\frac{\lambda}{2\pi x^3}\right]^{1/2} exp\left(\frac{-\lambda(x-\mu)^2}{2\mu^2 x}\right); x > 0,\ \mu > 0,\ \lambda > 0 \tag{1}$$

where $x$ is the independent variable, $\lambda$ represents the shape coefficient, and $\mu$ represents the expectation of the function. It has been confirmed that the characteristics of the inverse

Gaussian distribution function and delay distribution are very similar [14,15], and both the super-threshold region and the delay distribution of the sub-threshold region can be fitted very well. Figure 1a,b shows the results of fitting with the inverse Gaussian distribution under 0.6 V and 0.2 V, respectively, and Figure 2a,b is their corresponding cumulative distribution function curves (CDF). It can be found that the graph coincidence is very high, so this paper uses the IGD probability density function to predict the delay distribution curve, and the parameters of the function ($\mu$, $\lambda$) will be modeled with delay in the following section. In this article, the delay is represented by $Td$, and the correspondences are as follows:

$$\mu(Td) = \mu; \ \sigma^2(Td) = \frac{\mu^3}{\lambda} \tag{2}$$

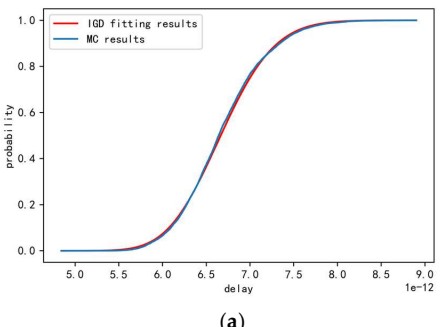

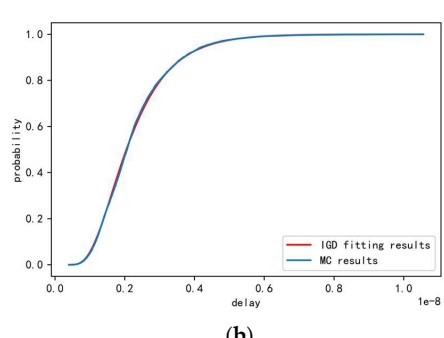

(a)                    (b)

**Figure 2.** CDF (**a**) 0.6 V (super-threshold); (**b**) 0.2 V (sub-threshold).

## 3. Delay Modeling

In this article, we consider variations in the threshold voltage as the main factor affecting the propagation delay, and the influence of other process variations can be translated into effective variations of the threshold voltage [3]. Therefore, the current selected in this article is a classic current expression containing the threshold voltage.

Drain–source current in the sub-threshold region for NMOS and PMOS [16] can be expressed as

$$I_n = I_{0n} \cdot e^{\frac{Vin(t)-Vthn-Vthb}{m_n V_T}} \cdot e^{\frac{\lambda_n Vds}{m_n V_T}} \cdot \left(1 - e^{-\frac{Vds}{V_T}}\right) \tag{3}$$

$$I_p = I_{0p} \cdot e^{\frac{(|Vin(t)|-|Vthp|-|Vthb|)}{m_p V_T}} \cdot e^{\frac{\lambda_p |Vds|}{m_p V_T}} \left(1 - e^{-\frac{|Vds|}{V_T}}\right) \tag{4}$$

where $I_{0n} = \mu_n C_{ox} \frac{W_n}{L_n}(m_n - 1)V_T^2$; $I_{0p} = \mu_p C_{ox} \frac{W_p}{l_p}(m_p - 1)V_T^2$, with the subscripts $n$ and $p$ here referring to NMOS and PMOS, respectively; $\mu$ is carrier mobility; $C_{ox}$ is gate oxide capacitance; $\frac{W}{L}$ is the width to length ratio; $m$ is sub-threshold slope; $V_T$ is thermal voltage; $\lambda$ represents the drain-induced barrier lowering (DIBL) effect coefficient; $Vth$ represents the threshold voltage at zero bias; and $Vthb$ is the increment of the threshold voltage caused by the body effect.

For inverters, taking the falling propagation delay ($Td$) as an example, it can be defined as the difference between the moment $t$ at which the output voltage drops to $Vdd/2$ and the time at which the operating voltage rises to $Vdd/2$, shown as

$$Td = t - \frac{\tau}{2} \tag{5}$$

where $\tau$ is the input transition time. At this time, it is necessary to consider the changes of the input voltage waveform and the output voltage waveform. Figure 3 depicts the input waveform (I) and the output waveform (II, III) curves. According to the size of the input transition time, it can be divided into fast input and slow input [17]. When the delay is greater than half of the input transition time, time $t0$, or when the output voltage drops to

$Vdd/2$ after the input transition time $\tau$, we consider this to be a case of fast input, as shown in Figure 3II, and vice versa in Figure 3III.

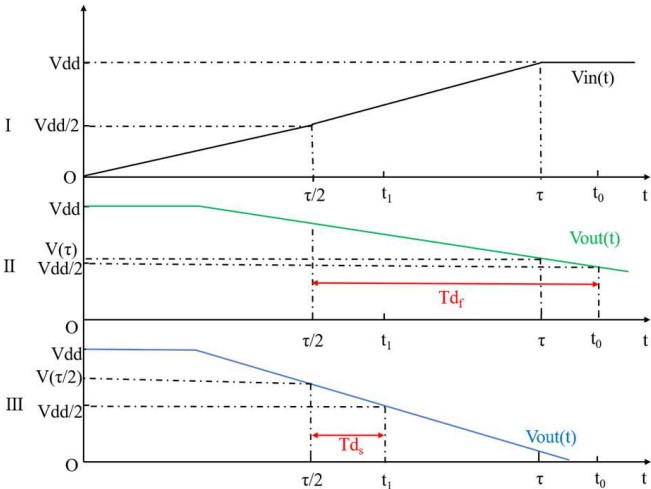

**Figure 3.** Input and output waveform (**I**: input waveform; **II**: output waveform under fast input condition; **III**: output waveform under slow input condition).

According to Figure 3, the input voltage can be expressed as

$$Vin(t) = \begin{cases} Vdd\frac{t}{\tau}, & 0 \leq t \leq \tau \\ Vdd, & t > \tau \end{cases} \tag{6}$$

The current can be re-expressed as [18]

$$I_{d0} = \mu_n C_{ox} \frac{W_n}{L_n}(m_n - 1)V_T^2 \cdot e^{\frac{\frac{t}{\tau}Vdd - Vthb - Vthn}{m_n V_T}} \cdot e^{\frac{\lambda_n Vds}{m_n V_T}}\left(1 - e^{-\frac{Vds}{V_T}}\right) \quad t < \tau \tag{7}$$

$$I_{d1} = \mu_n C_{ox} \frac{W_n}{L_n}(m_n - 1)V_T^2 \cdot e^{\frac{Vdd - |Vthb| - |Vthn|}{m_n V_T}} \cdot e^{\frac{\lambda_n Vds}{m_n V_T}}\left(1 - e^{-\frac{Vds}{V_T}}\right) \quad t > \tau \tag{8}$$

Next, we will derive the analytical expressions of the output voltage waveform and propagation delay for different situations.

### 3.1. Output Voltage Calculation

3.1.1. $0 \leq t \leq \tau$

According to Kirchhoff's current law (KCL) [19] at $Vout(t)$:

$$C_{tot}\frac{dVout(t)}{dt} = -I_{d0} \tag{9}$$

where $C_{tot}$ is the sum of the load capacitance and the coupling capacitance.

In this case, $Vgs = Vin(t) = \frac{Vdd}{\tau}t$, $Vds = Vout(t)$, and these are substituted into Equation (7) and the equation is phase shifted:

$$\frac{dVout(t)}{e^{\frac{\lambda_n Vout(t)}{m_n V_T}} \cdot \left(1 - e^{-\frac{Vout(t)}{V_T}}\right)} = \frac{-I_{0n} \cdot e^{\frac{-Vthb - Vthn}{m_n V_T}}}{C_L} \cdot e^{\frac{Vdd}{\tau}t}{m_n V_T}} \, dt \tag{10}$$

It is clear that this equation is unsolvable, so we have to use the approximation method to solve it. For inverters, an important point used to obtain the delay is when the output

voltage drops to $Vdd/2$, and when $Vout(t) = Vdd/2$, $e^{-\frac{Vout(t)}{V_T}}$ is small enough to be ignored, as shown in Equation (11):

$$\frac{dVout(t)}{e^{\frac{\lambda_n Vout(t)}{m_n V_T}}} = \frac{-I_{0n}\cdot e^{\frac{-Vthb-Vthn}{m_n V_T}}}{C_{tot}}\cdot e^{\frac{Vdd}{\tau} t}{m_n V_T}\; dt \tag{11}$$

By integrating both sides of the above equation and substituting the initial condition $Vout(t) = Vdd$ when $t = 0$, we can obtain the expression of $Vout(t)$ [18]:

$$Vout(t) = \frac{-m_n V_T}{\lambda_n}\cdot ln\left(\frac{I_{0n}\cdot e^{\frac{-Vthb-Vthn}{m_n V_T}}\lambda_n \tau}{Vdd\cdot C_{tot}}\left(e^{\frac{Vdd}{m_n V_T \tau}t} - 1\right) + e^{-\frac{\lambda_n Vdd}{m_n V_T}}\right) \tag{12}$$

At the same time, we can obtain $Vout(\tau)$:

$$Vout(\tau) = \frac{-m_n V_T}{\lambda_n}\cdot ln\left(\frac{I_{0n}\cdot e^{\frac{-Vthb-Vthn}{m_n V_T}}\lambda_n \tau}{Vdd\cdot C_{tot}}\left(e^{\frac{Vdd}{m_n V_T \tau}\tau} - 1\right) + e^{-\frac{\lambda_n Vdd}{m_n V_T}}\right) = \frac{-m_n V_T}{\lambda_n}\cdot ln\left(\frac{I_{0n}\cdot e^{\frac{-Vthb-Vthn}{m_n V_T}}\lambda_n \tau}{Vdd\cdot C_{tot}}\left(e^{\frac{Vdd}{m_n V_T}} - 1\right) + e^{-\frac{\lambda_n Vdd}{m_n V_T}}\right) \tag{13}$$

3.1.2. $t > \tau$

In this case, $dVin(t) = 0$, $Vin = Vdd$; substituting these into Equation (8) and phase shifting, then integrating and substituting the initial condition $Vout(t) = Vout(\tau)$ when $t = \tau$, we can obtain the expression:

$$Vout(t) = \frac{-m_n V_T}{\lambda_n}\cdot ln\left(\frac{I_{0n}\cdot e^{\frac{-Vthb-Vthn}{m_n V_T}}\lambda_n}{C_{tot} m_n V_T}e^{\frac{Vdd}{m_n V_T}}(t - \tau) + e^{-\frac{\lambda_n Vout(\tau)}{m_n V_T}}\right) \tag{14}$$

*3.2. Analytical Expression for Delay*

3.2.1. Fast Input

Fast input occurs in the case of $t > \tau$. By substituting Equation (14) into $Vout(t) = vdd/2$, we can obtain the time $t0$ at this time.

$$t0 = \frac{C_{tot} m_n V_T}{I_{0n}\cdot e^{\frac{-Vthb-Vthn}{m_n V_T}}\lambda_n}e^{\frac{-Vdd}{m_n V_T}}\left(e^{-\frac{\lambda_n Vdd}{2m_n V_T}} - e^{-\frac{\lambda_n Vout(\tau)}{m_n V_T}}\right) + \tau \tag{15}$$

According to the definition of delay, we can find its expression as follows:

$$Td = t0 - \frac{\tau}{2} = \frac{C_{tot} m_n V_T}{I_{0n}\cdot e^{\frac{-Vthb-Vthn}{m_n V_T}}\lambda_n}e^{\frac{-Vdd}{m_n V_T}}\left(e^{-\frac{\lambda_n Vdd}{2m_n V_T}} - e^{-\frac{\lambda_n Vout(\tau)}{m_n V_T}}\right) + \frac{\tau}{2} \tag{16}$$

In order to increase the applicability of this formula, we introduce a coefficient $k0$ of the number of samples in the MC simulation, which is performed later for MC verification; it can be simulated any number of times for verification. When the number of simulations is fixed, this coefficient is a constant, generally around 1. The new expression is

$$Td = k0\cdot \frac{C_{tot} m_n V_T}{I_{0n}\cdot e^{\frac{-Vthb-Vthn}{m_n V_T}}\lambda_n}e^{\frac{-Vdd}{m_n V_T}}\left(e^{-\frac{\lambda_n Vdd}{2m_n V_T}} - e^{-\frac{\lambda_n Vout(\tau)}{m_n V_T}}\right) + \frac{\tau}{2} \tag{17}$$

In order to find the variance of the delay, we need to sort out the above expression and combine the same influencing factors. The result is as follows:

$$Td = k0\cdot \frac{C_{tot} m_n V_T}{I_{0n}\cdot e^{\frac{-Vthb-Vthn}{m_n V_T}}\lambda_n}e^{\frac{-Vdd}{m_n V_T}}\left(e^{-\frac{\lambda_n Vdd}{2m_n V_T}} - e^{-\frac{\lambda_n Vdd}{m_n V_T}}\right) + \tau\cdot\left[\frac{1}{2} - k0\cdot \frac{m_n V_T}{Vdd}\cdot\left(1 - e^{\frac{-Vdd}{m_n V_T}}\right)\right] \tag{18}$$

Therefore, the variance $\sigma^2(Td)$ can be written as

$$\sigma^2(Td) = [\frac{k0 \cdot C_{tot} m_n V_T}{I_{0n} \lambda_n} e^{\frac{-Vdd}{m_n V_T}} \left(e^{-\frac{\lambda_n Vdd}{2m_n V_T}} - e^{-\frac{\lambda_n Vdd}{m_n V_T}}\right)]^2 \cdot \sigma^2\left(e^{\frac{Vthb+Vthn}{m_n V_T}}\right) + \left[\frac{1}{2} - \frac{k0 \cdot m_n V_T}{Vdd} \cdot \left(1 - e^{\frac{-Vdd}{m_n V_T}}\right)\right]^2 \cdot \sigma^2(\tau) \quad (19)$$

In fact, for the cell circuit, the change in input transition time does not have a large effect on the variance of the delay. Changes in the process parameters can be represented by variance changes in the threshold voltage, so the second half of the above expression can be removed to reduce the amount of calculation.

$$\sigma^2(Td) = [k0 \cdot \frac{C_{tot} m_n V_T}{I_{0n} \lambda_n} e^{\frac{-Vdd}{m_n V_T}} \left(e^{-\frac{\lambda_n Vdd}{2m_n V_T}} - e^{-\frac{\lambda_n Vdd}{m_n V_T}}\right)]^2 \cdot \sigma^2(e^{\frac{Vthb+Vthn}{m_n V_T}}) \quad (20)$$

3.2.2. Slow Input

Slow input occurs in the case of $0 \leq t \leq \tau$. By substituting Equation (12) into $Vout(t) = Vdd/2$, we can obtain the time $t1$ at this time.

$$Td = t1 - \frac{\tau}{2} = k0 \frac{m_n V_T \tau}{Vdd} ln[\frac{Vdd \cdot C_{tot}}{I_{0n} \cdot e^{\frac{-Vthb-Vthn}{m_n V_T}} \lambda_n \tau} (e^{\frac{-\lambda_n Vdd}{2m_n V_T}} - e^{-\frac{\lambda_n Vdd}{m_n V_T}}) + 1] - \frac{\tau}{2} \quad (21)$$

Similarly, solving the variance requires ignoring the effect of the input transition time on the propagation delay variance, and separating the threshold voltage [17], then we can obtain the following equation:

$$\sigma^2(Td) = \left(k0 \frac{\tau}{Vdd}\right)^2 \sigma^2(Vthb + Vthn) \quad (22)$$

For more details, please see 'Appendix B'.

## 4. Results and Discussion

Before verification, we must undertake preparation to obtain the values of various coefficients in the equations. First, when the temperature is certain, we need to calculate the value of the thermal voltage ($KT/q$). Then, we sweep $Vgs$ and $Vds$, respectively, from DC simulation, and use the ratio method to calculate the value of $m$ and $\lambda$. Next, we can obtain $I0$ by the least squares fitting method with the current data from DC. Finally, we can calculate $k0$ and the variance of the threshold voltage by a standard MC simulation. Although the MC simulation is used here, it is only performed once. Table 1 shows the method of obtaining each coefficient.

**Table 1.** Method of obtaining various parameters.

| Parameters | Method of Extraction or Calculation |
|---|---|
| $V_T$ | $V_T = KT/q$ |
| $Vth$ | DC simulation, with the command "$Vth$ (*)" |
| $m$ | DC simulation and ratio method |
| $\lambda$ | DC simulation and ratio method |
| $I0$ | DC simulation and least squares fitting method |
| $k0$ | MC simulation, calculated by a standard MC delay value |
| $\sigma^2 (Vth)$ | MC simulation |

In this article, the indicator used to calculate the mean and standard deviation of the delay is calculated as follows:

$$\sum_{1}^{n} \frac{|y - y0|}{y0} \quad (23)$$

where $y$ represents model prediction results, and $y0$ represents the simulation results with HSPICE.

For the distribution function, we generally use the difference between the piecewise integrals of the probability density function (PDF) to measure the error, and the cumulative distribution function (CDF) is the integral of the PDF; thus, the CDF can be used to calculate the indicator, shown as

$$\sum_{1}^{n} \frac{|CDF_{model} - CDF_{MC}|}{CDF_{MC}} \tag{24}$$

where $CDF_{model}$ represents the CDF of the model prediction results; $CDF_{MC}$ is the CDF of the simulation results with HSPICE; and n refers to the n-segment integration of the PDF. According to the classic value [15], here we take $n = 5$.

### 4.1. Current Verification

Fully depleted SOI (FDSOI) MOSFETs are ideal for low-power applications due to their superior control of short-channel effects and flexibility of dynamic threshold voltage through the use of back-gate bias [20,21]. In this paper, the model is validated in 22 nm FDSOI technology. First, a DC simulation is performed to obtain the coefficients required for the current; here, we set $Vbs = 0$. Additionally, Table 2 shows the corresponding coefficients at a temperature of $T = 25\ ^\circ$C; $\frac{W_n}{L_n} = 80$ nm/20 nm; $\frac{W_P}{L_P} = 235$ nm/20 nm. This set of coefficients only needs to be obtained once, and is further used in subsequent delay calculations.

**Table 2.** Coefficients from DC.

| Transistor | *Vth* (V) | $\lambda$ | *I0* (A) | *M* (mV/dec) | $V_T$ (V) |
|---|---|---|---|---|---|
| NMOS | 0.324 | 0.073 | $7.66 \times 10^{-7}$ | 1.462 | 0.0257 |
| PMOS | −0.325 | 0.093 | $7.11 \times 10^{-7}$ | 1.504 | 0.0257 |

The nominal value of the DC under different voltages is simulated and compared with the current value calculated by the current formula; the results show a high accuracy, with the error being less than 1%. Figure 4a,b is the current curves of NMOS and PMOS transistors under different *Vds* and *Vgs* voltages.

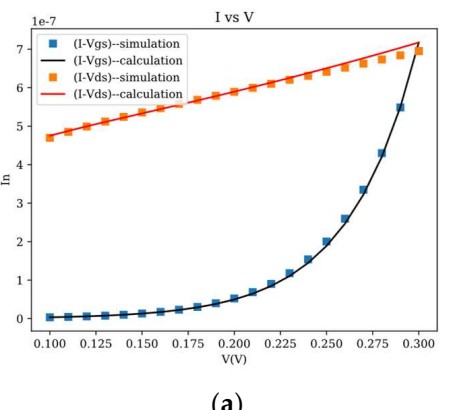
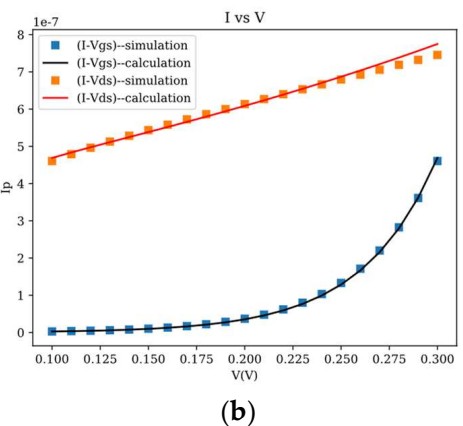

| (a) | (b) |

**Figure 4.** Current changes of transistors with voltage, where the symbols represent the DC simulation results and the lines represent the current formula calculation results: (**a**) NMOS; (**b**) PMOS.

The cell circuit in this paper uses an inverter, and its current is also verified with the above current formula and correlation coefficients. Figure 5a,b is the drain current curves of the inverter. It can be observed that it highly matches the standard current, indicating that a series of delay derivations using this current are feasible.

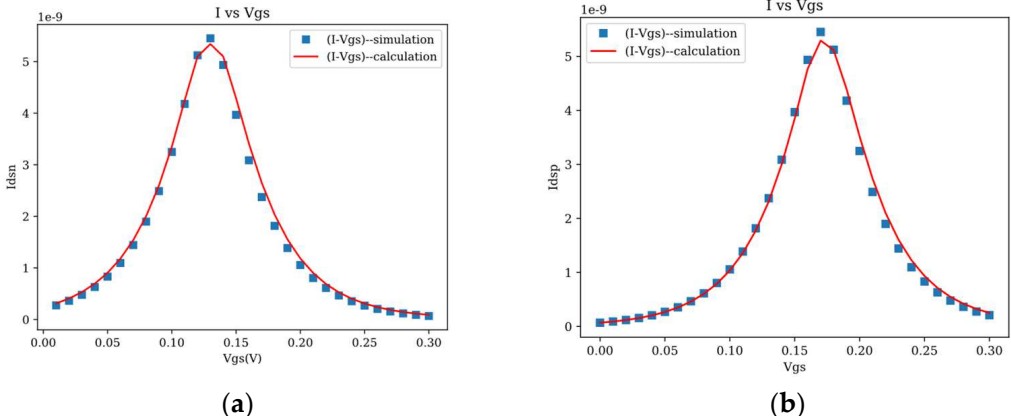

(**a**)                                                     (**b**)

**Figure 5.** Inverter current changes with voltage, where symbols represent the simulation results and the lines represent the current formula calculation results: (**a**) $I_n$; (**b**) $I_p$.

*4.2. Delay Verification*

Table 2 lists the coefficients required for the model. Before performing the delay verification, we must perform an MC simulation to obtain the $k0$ and variance $\sigma^2(Vthn)$.

In order to verify the accuracy of the proposed delay model and prediction method, the mean and standard deviation of the delay for an inverter with the process fluctuation parameter changes are simulated by SPICE using the 22 nm industrial design suite (the golden data are 10,000 samples of MC simulations), with results having a mean error of about 1.5% and a standard deviation error of about 4.3% in the sub-threshold region, indicating very high accuracy.

Figure 6a shows the prediction results of the delay under different voltages and different loads (Cl), and Figure 6b shows the prediction results of the standard deviation compared with MC simulation results.

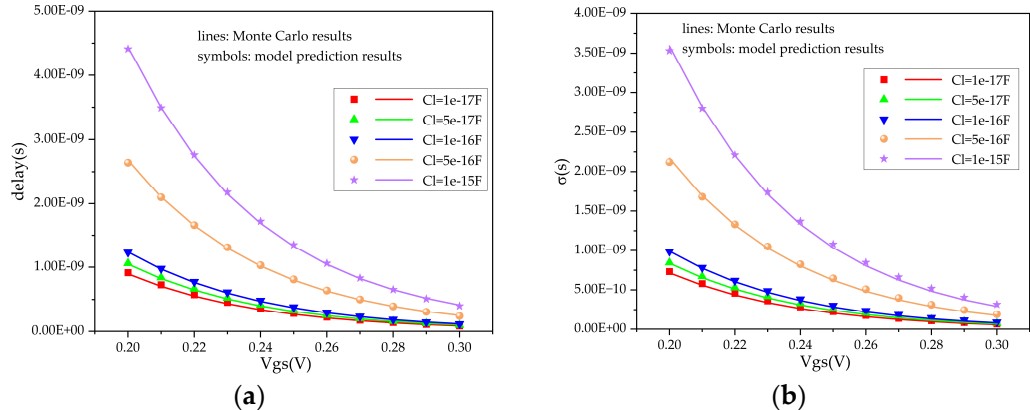

(**a**)                                                     (**b**)

**Figure 6.** Model prediction results of delay compared with the MC simulation results (T = 25 °C, $\tau = 1 \times 10^{-11}$ s) under different load capacitances and different voltages. (**a**) Mean of the delay; (**b**) standard deviation of the delay.

4.2.1. Transition Time Verification

We also validated the delay modeling of fast and slow inputs under fast input condition. Figure 7a shows the delay prediction results under different load conditions with different input transition times, and Figure 7b shows its standard deviation results.

In order to verify the correctness of the model, we randomly generate some multi-input values, and then predict the result of propagation delay. Table 3 shows part of the data. It can be seen that the error is acceptable for both the mean and standard deviation.

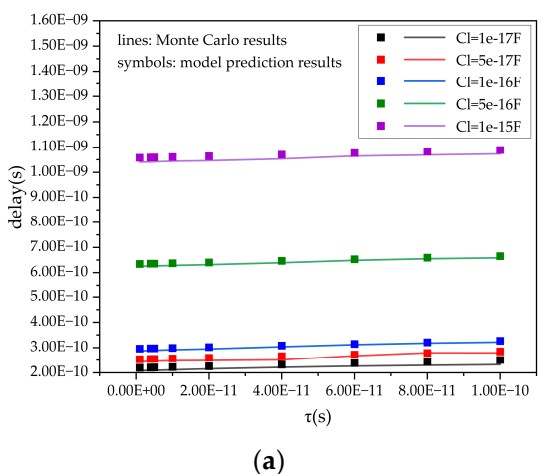 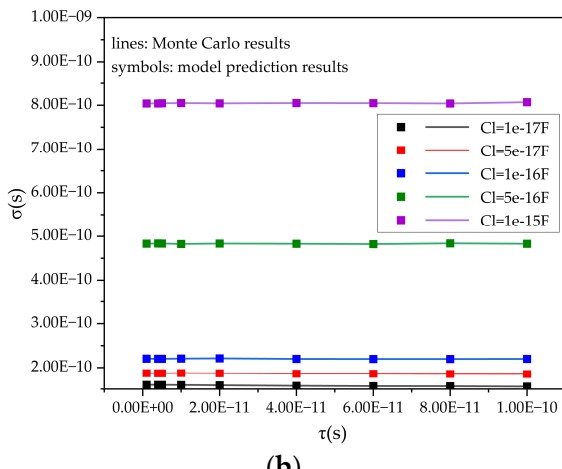

**Figure 7.** Fast input. Model prediction results of delay compared with the MC simulation results ($Vgs$ = 0.26 V) under different load capacitance and input transition times. (**a**) Mean of the delay; (**b**) standard deviation of the delay.

**Table 3.** Partial validation data under fast input condition.

| | | | $\mu$ (s) | | | $\sigma$ (s) | | |
|---|---|---|---|---|---|---|---|---|
| $Vgs$ (V) | Cl (F) | $\tau$ (s) | $\mu$ (MC) | $\mu$ (Model) | $\mu$ (Error) | $\sigma$ (MC) | $\sigma$ (Model) | $\sigma$ (Error) |
| 0.205 | $4.94 \times 10^{-16}$ | $1.79 \times 10^{-11}$ | $2.36 \times 10^{-9}$ | $2.39 \times 10^{-9}$ | 1.53% | $1.90 \times 10^{-9}$ | $1.88 \times 10^{-9}$ | 1.40% |
| 0.231 | $8.50 \times 10^{-17}$ | $3.17 \times 10^{-11}$ | $5.65 \times 10^{-10}$ | $5.94 \times 10^{-10}$ | 5.21% | $4.37 \times 10^{-10}$ | $4.59 \times 10^{-10}$ | 5.21% |
| 0.299 | $6.53 \times 10^{-16}$ | $6.11 \times 10^{-11}$ | $3.28 \times 10^{-10}$ | $3.20 \times 10^{-10}$ | 2.21% | $2.17 \times 10^{-10}$ | $2.36 \times 10^{-10}$ | 8.51% |
| 0.203 | $2.17 \times 10^{-16}$ | $6.40 \times 10^{-12}$ | $1.55 \times 10^{-9}$ | $1.58 \times 10^{-9}$ | 1.80% | $1.24 \times 10^{-9}$ | $1.24 \times 10^{-9}$ | 0.54% |
| 0.229 | $5.02 \times 10^{-16}$ | $1.90 \times 10^{-12}$ | $1.33 \times 10^{-9}$ | $1.37 \times 10^{-9}$ | 3.29% | $1.06 \times 10^{-9}$ | $1.08 \times 10^{-9}$ | 1.93% |
| 0.202 | $5.10 \times 10^{-17}$ | $6.55 \times 10^{-11}$ | $1.02 \times 10^{-9}$ | $1.05 \times 10^{-9}$ | 2.94% | $7.98 \times 10^{-10}$ | $8.17 \times 10^{-10}$ | 2.42% |
| 0.197 | $3.31 \times 10^{-16}$ | $1.02 \times 10^{-10}$ | $2.29 \times 10^{-9}$ | $2.27 \times 10^{-9}$ | 0.80% | $1.78 \times 10^{-9}$ | $1.77 \times 10^{-9}$ | 0.65% |
| 0.253 | $6.33 \times 10^{-16}$ | $4.65 \times 10^{-10}$ | $1.09 \times 10^{-9}$ | $1.05 \times 10^{-9}$ | 3.85% | $6.72 \times 10^{-10}$ | $7.12 \times 10^{-10}$ | 5.86% |
| 0.221 | $6.97 \times 10^{-16}$ | $5.83 \times 10^{-10}$ | $2.28 \times 10^{-9}$ | $2.25 \times 10^{-9}$ | 1.60% | $1.61 \times 10^{-9}$ | $1.64 \times 10^{-9}$ | 2.13% |
| 0.187 | $5.25 \times 10^{-16}$ | $1.89 \times 10^{-10}$ | $3.85 \times 10^{-9}$ | $3.81 \times 10^{-9}$ | 0.90% | $3.03 \times 10^{-9}$ | $2.96 \times 10^{-9}$ | 2.34% |

Additionally, Figure 8a,b shows the prediction results of mean and standard deviation in the case of the slow input condition.

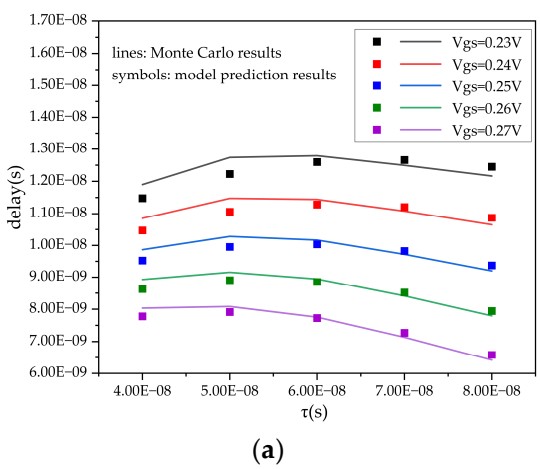 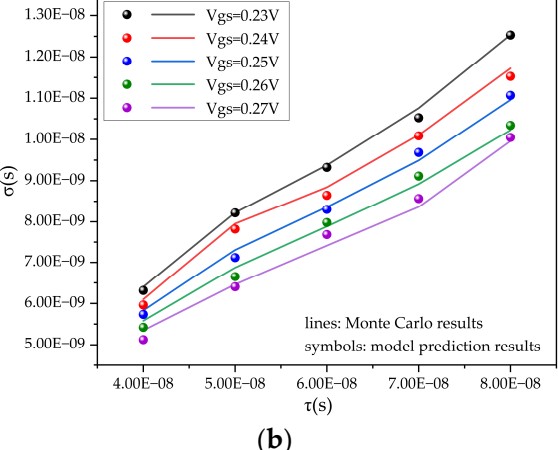

**Figure 8.** Model prediction results of delay compared with the MC simulation results (Cl = 1 fF) under different voltages and different input transition times. (**a**) Mean of the delay; (**b**) standard deviation of the delay.

Table 4 shows the comparison of the partial prediction data and the MC simulation results, as well as related errors under slow input conditions.

**Table 4.** Partial validation data under slow input conditions.

| | | | $\mu$ (s) | | | $\sigma$ (s) | | |
|---|---|---|---|---|---|---|---|---|
| *Vgs* (V) | Cl (F) | $\tau$ (s) | $\mu$ (MC) | $\mu$ (Model) | $\mu$ (Error) | $\sigma$ (MC) | $\sigma$ (Model) | $\sigma$ (Error) |
| 0.245 | $9.78 \times 10^{-16}$ | $8.68 \times 10^{-8}$ | $9.32 \times 10^{-9}$ | $9.43 \times 10^{-9}$ | 1.23% | $1.25 \times 10^{-8}$ | $1.23 \times 10^{-8}$ | 1.62% |
| 0.270 | $9.39 \times 10^{-16}$ | $7.46 \times 10^{-8}$ | $6.22 \times 10^{-9}$ | $6.36 \times 10^{-9}$ | 2.25% | $8.80 \times 10^{-9}$ | $9.56 \times 10^{-9}$ | 8.63% |
| 0.299 | $1.01 \times 10^{-15}$ | $4.00 \times 10^{-8}$ | $5.52 \times 10^{-9}$ | $5.48 \times 10^{-9}$ | 0.73% | $4.65 \times 10^{-9}$ | $4.61 \times 10^{-9}$ | 0.86% |
| 0.293 | $8.42 \times 10^{-16}$ | $4.27 \times 10^{-8}$ | $5.11 \times 10^{-9}$ | $5.08 \times 10^{-9}$ | 0.72% | $4.96 \times 10^{-9}$ | $5.04 \times 10^{-9}$ | 1.75% |
| 0.279 | $5.68 \times 10^{-16}$ | $4.98 \times 10^{-8}$ | $3.78 \times 10^{-9}$ | $3.62 \times 10^{-9}$ | 4.23% | $5.75 \times 10^{-8}$ | $6.17 \times 10^{-8}$ | 7.30% |
| 0.290 | $2.01 \times 10^{-15}$ | $5.00 \times 10^{-8}$ | $1.08 \times 10^{-8}$ | $1.06 \times 10^{-8}$ | 1.85% | $6.14 \times 10^{-9}$ | $5.96 \times 10^{-9}$ | 2.93% |
| 0.261 | $5.12 \times 10^{-16}$ | $3.79 \times 10^{-8}$ | $5.24 \times 10^{-9}$ | $5.15 \times 10^{-9}$ | 1.61% | $5.18 \times 10^{-9}$ | $5.03 \times 10^{-9}$ | 2.98% |
| 0.223 | $5.30 \times 10^{-16}$ | $2.47 \times 10^{-8}$ | $7.57 \times 10^{-9}$ | $7.49 \times 10^{-9}$ | 1.04% | $4.06 \times 10^{-9}$ | $3.83 \times 10^{-9}$ | 5.84% |
| 0.24 | $6.81 \times 10^{-16}$ | $3.45 \times 10^{-8}$ | $8.28 \times 10^{-9}$ | $8.04 \times 10^{-9}$ | 2.95% | $5.28 \times 10^{-9}$ | $4.97 \times 10^{-9}$ | 5.83% |
| 0.287 | $7.00 \times 10^{-16}$ | $4.06 \times 10^{-8}$ | $4.73 \times 10^{-9}$ | $4.67 \times 10^{-9}$ | 1.31% | $4.89 \times 10^{-9}$ | $4.89 \times 10^{-9}$ | 0.06% |

### 4.2.2. Verification in Different Temperatures

All the above verification was performed at a temperature of 25 °C, though the formula we propose is also very accurate at other temperatures. When the temperature changes, we only need to reperform a DC simulation on the MOS transistor to obtain the coefficients, as well as a MC simulation to obtain $\sigma^2(Vthn)$ for different temperatures, and then update them. Table 5 lists the coefficients at different temperatures for NMOS. Finally, by substituting the coefficients in the expressions, we can obtain the results under different inputs.

**Table 5.** Coefficients at different temperatures for NMOS.

| T (°C) | Vth (V) | $\lambda$ | I0 (A) | $m$ (mV/dec) | $V_T$ (V) |
|---|---|---|---|---|---|
| −40 | 0.364 | 0.072 | $7.82 \times 10^{-7}$ | 1.410 | 0.0200 |
| 0 | 0.340 | 0.072 | $7.82 \times 10^{-7}$ | 1.435 | 0.0235 |
| 50 | 0.308 | 0.076 | $7.42 \times 10^{-7}$ | 1.487 | 0.0278 |
| 100 | 0.273 | 0.088 | $6.87 \times 10^{-7}$ | 1.567 | 0.0322 |
| 125 | 0.254 | 0.098 | $6.62 \times 10^{-7}$ | 1.617 | 0.0343 |

In order to further verify the feasibility of our proposed model, we carried out corresponding experiments at different temperatures. Table 4 shows that the threshold voltage is 0.254 V at 125 °C, and this paper studies the delay model in the sub-threshold region, so the voltage selection in Figure 9 is lower than 0.25 V. It is obvious that the model results and the results of the SPICE MC simulation are highly matched. Table 6 lists the average errors at different temperatures, all within the acceptable range.

**Table 6.** Average errors at different temperatures.

| T (°C) | −40 | 0 | 50 | 100 | 125 |
|---|---|---|---|---|---|
| $\mu$ (error) | 0.75% | 1.83% | 2.30% | 2.98% | 3.18% |
| $\sigma$ (error) | 1.67% | 3.12% | 3.15% | 4.24% | 4.21% |

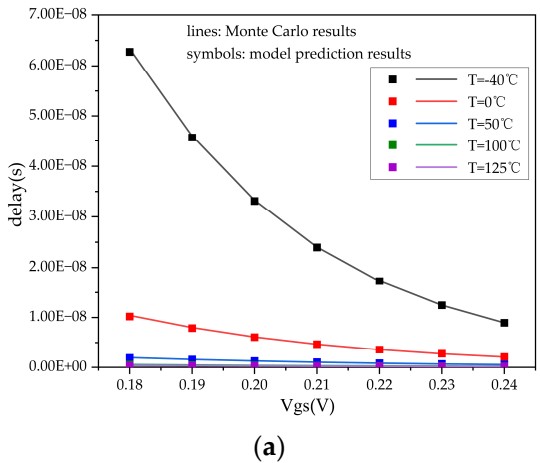 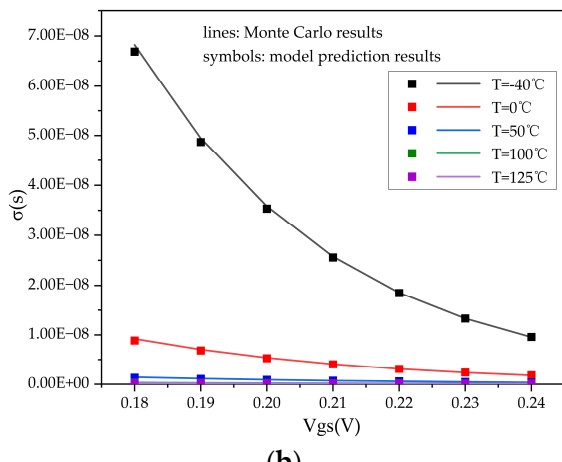

(**a**)　　　　　　　　　　　　　　　　　　　(**b**)

**Figure 9.** Model prediction results compared with the MC simulation results (Cl = 0.5 fF, $\tau = 5 \times 10^{-11}$ s) for an inverter under different voltages and temperatures. (**a**) Mean of the delay; (**b**) standard deviation of the delay.

### 4.2.3. Verification for Different Transistor Sizes

In the verification above, the length of the transistor we choose was 20 nm. In fact, our model is suitable for different sizes. We also selected for sizes to verify their accuracy. Details of the four sizes are shown in Table 7.

**Table 7.** Coefficients for NMOS in the case of different sizes.

| Size | W (nm) | L (nm) | Vth (V) | $\lambda$ | I0 (A) | m (mV/dec) | $V_T$ (V) |
|---|---|---|---|---|---|---|---|
| Size 1 | 80 | 30 | 0.340 | 0.042 | $6.58 \times 10^{-7}$ | 1.285 | 0.0257 |
| Size 2 | 100 | 40 | 0.347 | 0.030 | $6.98 \times 10^{-7}$ | 1.218 | 0.0257 |
| Size 3 | 135 | 50 | 0.352 | 0.024 | $8.10 \times 10^{-7}$ | 1.185 | 0.0257 |
| Size 4 | 150 | 60 | 0.356 | 0.019 | $7.86 \times 10^{-7}$ | 1.165 | 0.0257 |

The current data and related errors of each size are shown in Appendix A. More details please see Tables A1–A5. Additionally, Figure 10 shows the delay results compared with MC simulation results under different operating voltages.

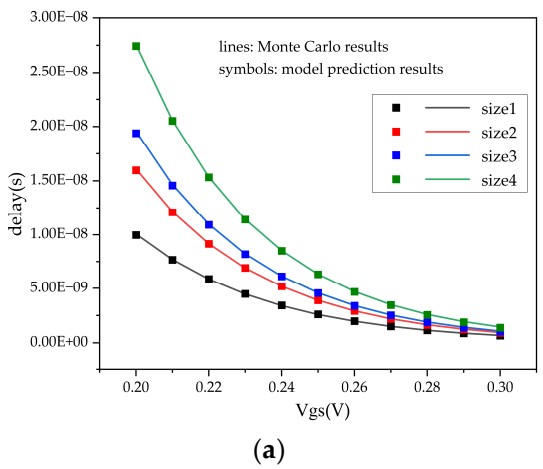 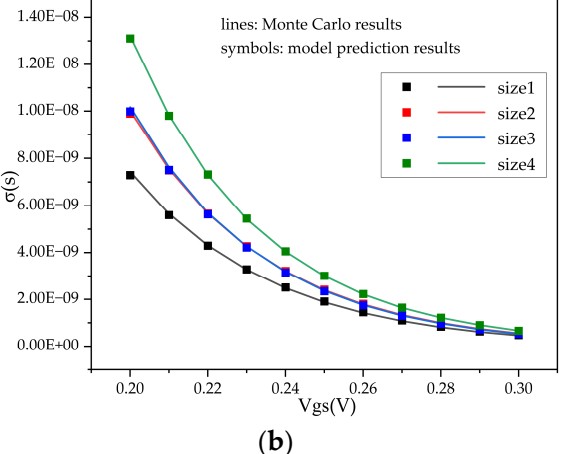

(**a**)　　　　　　　　　　　　　　　　　　　(**b**)

**Figure 10.** Model prediction results compared with the MC simulation results (Cl = 0.5 fF, $\tau = 1 \times 10^{-11}$ s, T = 25 °C) for an inverter under different voltages and sizes. (**a**) Mean of the delay; (**b**) standard deviation of the delay.

Moreover, we tested a significant amount of data for each size, and Tables 8–11 show the partial data and corresponding errors. The average error of each size is less than 5%.

**Table 8.** Size 1: Partial validation data (T = 25 °C).

| | | | $\mu$ (s) | | | $\sigma$ (s) | | |
|---|---|---|---|---|---|---|---|---|
| Vgs (V) | Cl (F) | $\tau$ (s) | $\mu$ (MC) | $\mu$ (Model) | $\mu$ (Error) | $\sigma$ (MC) | $\sigma$ (Model) | $\sigma$ (Error) |
| 0.238 | $4.81 \times 10^{-16}$ | $4.45 \times 10^{-10}$ | $3.650440 \times 10^{-9}$ | $3.676056 \times 10^{-9}$ | 0.70% | $2.551130 \times 10^{-9}$ | $2.587672 \times 10^{-9}$ | 1.43% |
| 0.243 | $2.38 \times 10^{-16}$ | $4.39 \times 10^{-10}$ | $2.340220 \times 10^{-9}$ | $2.334835 \times 10^{-9}$ | 0.23% | $1.588780 \times 10^{-9}$ | $1.606478 \times 10^{-9}$ | 1.11% |
| 0.289 | $5.40 \times 10^{-16}$ | $1.70 \times 10^{-10}$ | $9.945210 \times 10^{-10}$ | $9.927272 \times 10^{-10}$ | 0.18% | $6.561770 \times 10^{-10}$ | $6.832115 \times 10^{-10}$ | 4.12% |
| 0.209 | $5.14 \times 10^{-16}$ | $2.39 \times 10^{-10}$ | $8.066620 \times 10^{-9}$ | $8.056563 \times 10^{-9}$ | 0.12% | $5.891210 \times 10^{-9}$ | $5.845090 \times 10^{-9}$ | 0.78% |
| 0.273 | $3.26 \times 10^{-16}$ | $4.43 \times 10^{-10}$ | $1.261890 \times 10^{-9}$ | $1.257850 \times 10^{-9}$ | 0.32% | $7.702130 \times 10^{-10}$ | $8.108863 \times 10^{-10}$ | 5.28% |
| 0.298 | $7.45 \times 10^{-16}$ | $3.18 \times 10^{-10}$ | $1.026480 \times 10^{-9}$ | $1.004063 \times 10^{-9}$ | 2.18% | $6.163550 \times 10^{-10}$ | $6.529836 \times 10^{-10}$ | 5.94% |
| 0.266 | $5.83 \times 10^{-16}$ | $4.71 \times 10^{-10}$ | $1.989130 \times 10^{-9}$ | $2.004680 \times 10^{-9}$ | 0.78% | $1.299260 \times 10^{-9}$ | $1.351936 \times 10^{-9}$ | 4.05% |
| 0.293 | $1.10 \times 10^{-16}$ | $1.89 \times 10^{-11}$ | $4.622390 \times 10^{-10}$ | $4.425993 \times 10^{-10}$ | 4.25% | $3.147220 \times 10^{-10}$ | $3.190541 \times 10^{-10}$ | 1.38% |
| 0.292 | $1.24 \times 10^{-16}$ | $8.46 \times 10^{-11}$ | $4.889020 \times 10^{-10}$ | $4.911607 \times 10^{-10}$ | 0.46% | $3.284180 \times 10^{-10}$ | $3.378408 \times 10^{-10}$ | 2.87% |
| 0.248 | $7.04 \times 10^{-16}$ | $1.52 \times 10^{-10}$ | $3.398730 \times 10^{-9}$ | $3.459904 \times 10^{-9}$ | 1.80% | $2.432340 \times 10^{-9}$ | $2.496145 \times 10^{-9}$ | 2.62% |

**Table 9.** Size 2: Partial validation data (T = 25 °C).

| | | | $\mu$ (s) | | | $\sigma$ (s) | | |
|---|---|---|---|---|---|---|---|---|
| Vgs (V) | Cl (F) | $\tau$ (s) | $\mu$ (MC) | $\mu$ (Model) | $\mu$ (Error) | $\sigma$ (MC) | $\sigma$ (Model) | $\sigma$ (Error) |
| 0.29 | $8.600 \times 10^{-17}$ | $3.390 \times 10^{-10}$ | $8.099090 \times 10^{-10}$ | $7.998129 \times 10^{-10}$ | 1.25% | $3.882130 \times 10^{-10}$ | $4.194068 \times 10^{-10}$ | 8.04% |
| 0.261 | $6.230 \times 10^{-16}$ | $4.090 \times 10^{-11}$ | $3.213440 \times 10^{-9}$ | $3.233015 \times 10^{-9}$ | 0.61% | $1.961680 \times 10^{-9}$ | $1.994106 \times 10^{-9}$ | 1.65% |
| 0.234 | $5.980 \times 10^{-16}$ | $4.787 \times 10^{-10}$ | $6.902490 \times 10^{-9}$ | $6.962671 \times 10^{-9}$ | 0.87% | $4.153020 \times 10^{-9}$ | $4.215779 \times 10^{-9}$ | 1.51% |
| 0.282 | $3.510 \times 10^{-16}$ | $9.800 \times 10^{-12}$ | $1.327440 \times 10^{-9}$ | $1.300852 \times 10^{-9}$ | 2.00% | $7.941080 \times 10^{-10}$ | $8.037207 \times 10^{-10}$ | 1.21% |
| 0.249 | $5.290 \times 10^{-16}$ | $2.362 \times 10^{-10}$ | $4.191040 \times 10^{-9}$ | $4.217566 \times 10^{-9}$ | 0.63% | $2.521270 \times 10^{-9}$ | $2.563139 \times 10^{-9}$ | 1.66% |
| 0.274 | $4.890 \times 10^{-16}$ | $4.695 \times 10^{-10}$ | $2.089330 \times 10^{-9}$ | $2.094021 \times 10^{-9}$ | 0.22% | $1.159100 \times 10^{-9}$ | $1.194233 \times 10^{-9}$ | 3.03% |
| 0.271 | $2.160 \times 10^{-16}$ | $3.918 \times 10^{-10}$ | $1.613750 \times 10^{-9}$ | $1.613682 \times 10^{-9}$ | 0.00% | $8.894820 \times 10^{-10}$ | $9.140999 \times 10^{-10}$ | 2.77% |
| 0.223 | $5.730 \times 10^{-16}$ | $3.074 \times 10^{-10}$ | $9.105560 \times 10^{-9}$ | $9.160867 \times 10^{-9}$ | 0.61% | $5.574670 \times 10^{-9}$ | $5.614147 \times 10^{-9}$ | 0.71% |
| 0.281 | $4.710 \times 10^{-16}$ | $2.773 \times 10^{-10}$ | $1.659610 \times 10^{-9}$ | $1.641169 \times 10^{-9}$ | 1.11% | $9.290240 \times 10^{-10}$ | $9.552446 \times 10^{-10}$ | 2.82% |
| 0.232 | $4.840 \times 10^{-16}$ | $2.979 \times 10^{-10}$ | $6.470420 \times 10^{-9}$ | $6.498238 \times 10^{-9}$ | 0.43% | $3.939820 \times 10^{-9}$ | $3.965225 \times 10^{-9}$ | 0.64% |

**Table 10.** Size3: Partial validation data (T = 25 °C).

| | | | $\mu$ (s) | | | $\sigma$ (s) | | |
|---|---|---|---|---|---|---|---|---|
| Vgs (V) | Cl (F) | $\tau$ (s) | $\mu$ (MC) | $\mu$ (Model) | $\mu$ (Error) | $\sigma$ (MC) | $\sigma$ (Model) | $\sigma$ (Error) |
| 0.214 | $2.71 \times 10^{-16}$ | $1.67 \times 10^{-10}$ | $1.016530 \times 10^{-8}$ | $1.014256 \times 10^{-8}$ | 0.22% | $5.286070 \times 10^{-9}$ | $5.198566 \times 10^{-9}$ | 1.66% |
| 0.244 | $4.19 \times 10^{-16}$ | $2.71 \times 10^{-10}$ | $5.089810 \times 10^{-9}$ | $5.099005 \times 10^{-9}$ | 0.18% | $2.580310 \times 10^{-9}$ | $2.578634 \times 10^{-9}$ | 0.06% |
| 0.278 | $5.62 \times 10^{-16}$ | $3.14 \times 10^{-10}$ | $2.261600 \times 10^{-9}$ | $2.222832 \times 10^{-9}$ | 1.71% | $1.069950 \times 10^{-9}$ | $1.085720 \times 10^{-9}$ | 1.47% |
| 0.28 | $7.01 \times 10^{-16}$ | $1.25 \times 10^{-10}$ | $2.332220 \times 10^{-9}$ | $2.284576 \times 10^{-9}$ | 2.04% | $1.137260 \times 10^{-9}$ | $1.153580 \times 10^{-9}$ | 1.43% |
| 0.262 | $7.68 \times 10^{-16}$ | $6.21 \times 10^{-11}$ | $4.060420 \times 10^{-9}$ | $4.054113 \times 10^{-9}$ | 0.16% | $2.084980 \times 10^{-9}$ | $2.077758 \times 10^{-9}$ | 0.35% |
| 0.232 | $6.82 \times 10^{-16}$ | $2.94 \times 10^{-10}$ | $9.182170 \times 10^{-9}$ | $9.185360 \times 10^{-9}$ | 0.03% | $4.690660 \times 10^{-9}$ | $4.681673 \times 10^{-9}$ | 0.19% |
| 0.256 | $9.89 \times 10^{-16}$ | $2.12 \times 10^{-10}$ | $5.728750 \times 10^{-9}$ | $5.712757 \times 10^{-9}$ | 0.28% | $2.879840 \times 10^{-9}$ | $2.905023 \times 10^{-9}$ | 0.87% |
| 0.267 | $6.46 \times 10^{-16}$ | $4.15 \times 10^{-10}$ | $3.328950 \times 10^{-9}$ | $3.299695 \times 10^{-9}$ | 0.88% | $1.598810 \times 10^{-9}$ | $1.622513 \times 10^{-9}$ | 1.48% |
| 0.271 | $2.16 \times 10^{-16}$ | $3.92 \times 10^{-10}$ | $1.902670 \times 10^{-9}$ | $1.912940 \times 10^{-9}$ | 0.54% | $8.971570 \times 10^{-10}$ | $9.117871 \times 10^{-10}$ | 1.63% |
| 0.252 | $1.97 \times 10^{-16}$ | $3.04 \times 10^{-10}$ | $3.124600 \times 10^{-9}$ | $3.133580 \times 10^{-9}$ | 0.29% | $1.555950 \times 10^{-9}$ | $1.559070 \times 10^{-9}$ | 0.20% |

**Table 11.** Size4: Partial validation data (T = 25 °C).

| Vgs (V) | Cl (F) | τ (s) | μ (MC) | μ (Model) | μ (Error) | σ (MC) | σ (Model) | σ (Error) |
|---|---|---|---|---|---|---|---|---|
| | | | $\mu$ (s) | | | $\sigma$ (s) | | |
| 0.294 | $6.71 \times 10^{-16}$ | $1.49 \times 10^{-10}$ | $2.054960 \times 10^{-9}$ | $1.983820 \times 10^{-9}$ | 3.46% | $9.274530 \times 10^{-10}$ | $9.209276 \times 10^{-10}$ | 0.70% |
| 0.276 | $9.05 \times 10^{-16}$ | $2.18 \times 10^{-10}$ | $4.056030 \times 10^{-9}$ | $4.028091 \times 10^{-9}$ | 0.69% | $1.882430 \times 10^{-9}$ | $1.885995 \times 10^{-9}$ | 0.19% |
| 0.236 | $7.28 \times 10^{-16}$ | $2.54 \times 10^{-10}$ | $1.155630 \times 10^{-8}$ | $1.163491 \times 10^{-8}$ | 0.68% | $5.442750 \times 10^{-9}$ | $5.516690 \times 10^{-9}$ | 1.36% |
| 0.206 | $1.82 \times 10^{-16}$ | $7.06 \times 10^{-11}$ | $1.661170 \times 10^{-8}$ | $1.636127 \times 10^{-8}$ | 1.51% | $7.951180 \times 10^{-9}$ | $7.806130 \times 10^{-9}$ | 1.82% |
| 0.259 | $3.63 \times 10^{-16}$ | $7.78 \times 10^{-11}$ | $4.246820 \times 10^{-9}$ | $4.232936 \times 10^{-9}$ | 0.33% | $1.996720 \times 10^{-9}$ | $2.008986 \times 10^{-9}$ | 0.61% |
| 0.228 | $2.10 \times 10^{-16}$ | $2.51 \times 10^{-10}$ | $9.044010 \times 10^{-9}$ | $8.975501 \times 10^{-9}$ | 0.76% | $4.288210 \times 10^{-9}$ | $4.247207 \times 10^{-9}$ | 0.96% |
| 0.237 | $3.64 \times 10^{-16}$ | $3.49 \times 10^{-10}$ | $8.247390 \times 10^{-9}$ | $8.240120 \times 10^{-9}$ | 0.09% | $3.871480 \times 10^{-9}$ | $3.878882 \times 10^{-9}$ | 0.19% |
| 0.285 | $4.47 \times 10^{-16}$ | $2.89 \times 10^{-10}$ | $2.243840 \times 10^{-9}$ | $2.193539 \times 10^{-9}$ | 2.24% | $9.737410 \times 10^{-10}$ | $9.963785 \times 10^{-10}$ | 2.32% |
| 0.216 | $5.37 \times 10^{-16}$ | $3.79 \times 10^{-10}$ | $1.786640 \times 10^{-8}$ | $1.793034 \times 10^{-8}$ | 0.36% | $8.495440 \times 10^{-9}$ | $8.506323 \times 10^{-9}$ | 0.13% |
| 0.26 | $8.36 \times 10^{-16}$ | $1.87 \times 10^{-10}$ | $6.156130 \times 10^{-9}$ | $6.168514 \times 10^{-9}$ | 0.20% | $2.832280 \times 10^{-9}$ | $2.914898 \times 10^{-9}$ | 2.92% |

### 4.2.4. Verification of Different Gates

The above results were verified with an inverter and achieved a high degree of accuracy. In fact, the model derived in this article is also applicable to other cell circuits. Here, we select the NAND2 gate ($\frac{W_n}{L_n} = 80$ nm/20 nm; $\frac{W_P}{L_P} = 110$ nm/20 nm) and NOR2 gate ($\frac{W_n}{L_n} = 80$ nm/20 nm; $\frac{W_P}{L_P} = 310$nm/20nm). Similarly, we first operate a DC for each transistor and the coefficients are shown in Table 12. Then, we verify the delay at different operating voltages, and Figure 11a,b shows the prediction results of the mean and standard deviation, respectively.

**Table 12.** Coefficients from DC (T = 25 °C).

| Gate | Transistor | Vth (V) | λ | I0 (A) | m (mV/dec) | $V_T$ (V) |
|---|---|---|---|---|---|---|
| NAND2 | NMOS | 0.324 | 0.073 | $7.66 \times 10^{-7}$ | 1.462 | 0.0257 |
| | PMOS | −0.325 | 0.089 | $3.49 \times 10^{-7}$ | 1.451 | 0.0257 |
| NOR2 | NMOS | 0.324 | 0.073 | $7.66 \times 10^{-7}$ | 1.462 | 0.0257 |
| | PMOS | −0.325 | 0.094 | $9.90 \times 10^{-7}$ | 1.517 | 0.0257 |

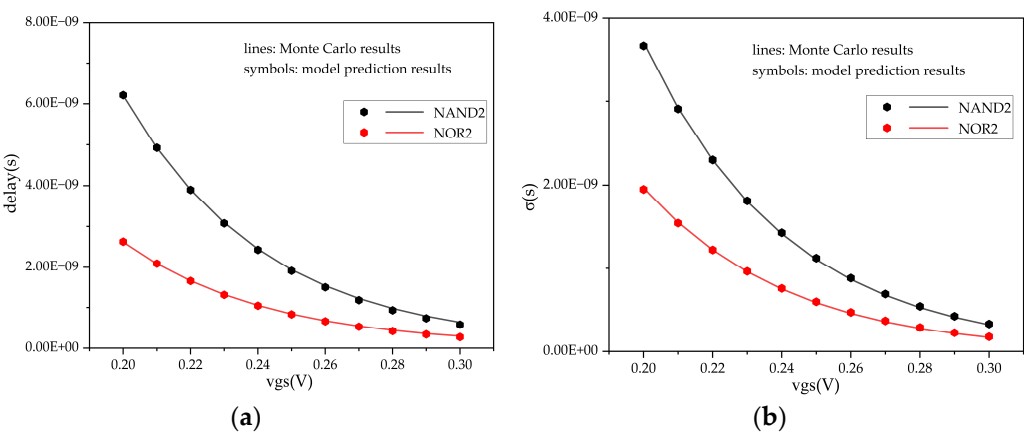

(a)      (b)

**Figure 11.** Model prediction results compared with the MC simulation results (T = 25 °C, Cl = 0.5 fF, $\tau = 1 \times 10^{-10}$ s) under different voltages. (**a**) Mean of the delay; (**b**) standard deviation of the delay.

From Figure 11, we can clearly see that the delay prediction results for each gate are very close to the MC simulation results. Additionally, the corresponding errors for each gate are shown in Table 13.

**Table 13.** Average errors of NAND2 and NOR2 gates.

| Gate | $\mu$ (Error) | $\sigma$ (Error) |
|---|---|---|
| NAND2 | 2.73% | 1.59% |
| NOR2 | 2.29% | 2.08% |

### 4.2.5. Verification Considering the Body Effect

In the above validation, we set $Vbs = 0$. In fact, the effect of the body effect on propagation delay is also modeled in this paper. It mainly affects the threshold voltage, and in Equation (7), $Vthb$ is the increase in the threshold voltage caused by the body effect.

Considering $Vbs$, we verify the delay for an inverter in the case of $|Vbs| > 0$, and Table 14 shows the coefficients obtained by DC simulations for NMOS.

**Table 14.** Coefficients at different $|Vbs|$ for NMOS (T = 25 °C).

| $|Vbs|$ (V) | $Vth$ (V) | $\lambda$ | $I0$ (A) | m (mV/dec) | $V_T$ (V) |
|---|---|---|---|---|---|
| 0.05 | 0.327 | 0.073 | $7.72 \times 10^{-7}$ | 1.453 | 0.0257 |
| 0.10 | 0.330 | 0.073 | $7.78 \times 10^{-7}$ | 1.449 | 0.0257 |
| 0.15 | 0.333 | 0.074 | $7.84 \times 10^{-7}$ | 1.444 | 0.0257 |

Additionally, the Figure 12a,b show the prediction results of the propagation delay compared with MC simulations under different voltages, and the error of the mean and standard deviation is about 1.4% and 3.2%, respectively.

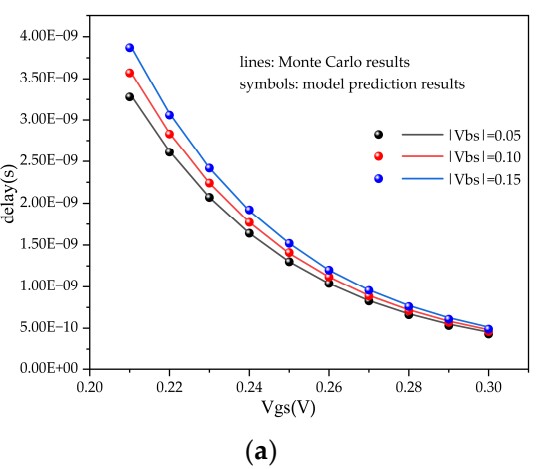

**(a)**

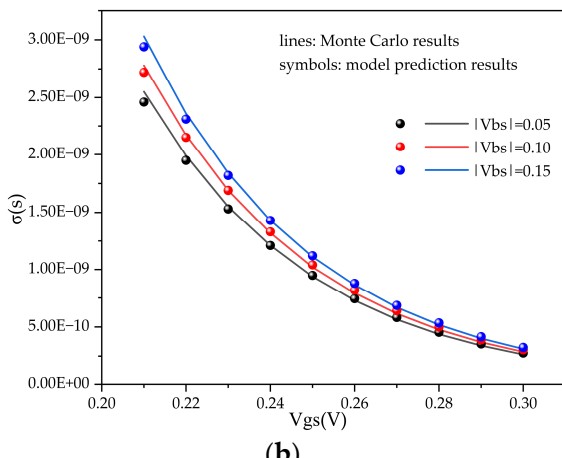

**(b)**

**Figure 12.** Model prediction results compared with the MC simulation results (T = 25 °C, Cl = 0.7 fF, $\tau = 2 \times 10^{-10}$ s) considering the body effect under different voltages. (**a**) Mean of the delay; (**b**) standard deviation of the delay.

### 4.2.6. Verification under Different Technologies

The above results were verified under 22 nm technology and achieved a high degree of accuracy. To further prove the universality of our model, we also validated it under another two technologies: 28 nm CMOS and 40 nm CMOS technologies. For an inverter, the size we selected is as follows: $\frac{W_n}{L_n} = 100$ nm/30 nm, $\frac{W_P}{L_P} = 200$ nm/30 nm for 28 nm technology, and $\frac{W_n}{L_n} = 120$ nm/40 nm, $\frac{W_P}{L_P} = 240$ nm/40 nm for 40 nm technology. The verification process is the same as that in the 22 nm technology.

The Table 15 displays the corresponding parameters for each technology.

**Table 15.** Coefficients under different technologies for NMOS.

| Technology | $Vth$ (V) | $\lambda$ | $I0$ (A) | m (mV/dec) | $V_T$ (V) |
|---|---|---|---|---|---|
| 28 nm | 0.374 | 0.149 | $5.98 \times 10^{-7}$ | 1.665 | 0.0257 |
| 40 nm | 0.597 | 0.121 | $1.32 \times 10^{-6}$ | 1.472 | 0.0257 |

Additionally, for multi-inputs, we predicted the propagation delay and the standard deviation for each technology; all the average errors are less than 4%. Additionally, Tables 16 and 17 show the partial data compared to the MC results.

**Table 16.** Partial data under 28 nm technology (T = 25 °C, $Vbs = 0$).

| | | | $\mu$ (s) | | | $\sigma$ (s) | | |
|---|---|---|---|---|---|---|---|---|
| $Vgs$ (V) | Cl (F) | $\tau$ (s) | $\mu$ (MC) | $\mu$ (Model) | $\mu$ (Error) | $\sigma$ (MC) | $\sigma$ (Model) | $\sigma$ (Error) |
| 0.278 | $9.00 \times 10^{-17}$ | $1.71 \times 10^{-10}$ | $6.944630 \times 10^{-10}$ | $6.838224 \times 10^{-10}$ | 1.53% | $7.639800 \times 10^{-10}$ | $7.839069 \times 10^{-10}$ | 2.61% |
| 0.268 | $2.96 \times 10^{-16}$ | $1.40 \times 10^{-10}$ | $1.284680 \times 10^{-9}$ | $1.290465 \times 10^{-9}$ | 0.45% | $1.553410 \times 10^{-9}$ | $1.535944 \times 10^{-9}$ | 1.12% |
| 0.259 | $3.84 \times 10^{-16}$ | $1.87 \times 10^{-10}$ | $1.804810 \times 10^{-9}$ | $1.814650 \times 10^{-9}$ | 0.55% | $2.250080 \times 10^{-9}$ | $2.164636 \times 10^{-9}$ | 3.80% |
| 0.291 | $2.43 \times 10^{-16}$ | $5.34 \times 10^{-11}$ | $6.980870 \times 10^{-10}$ | $6.957877 \times 10^{-10}$ | 0.33% | $7.894610 \times 10^{-10}$ | $8.330983 \times 10^{-10}$ | 5.53% |
| 0.267 | $1.57 \times 10^{-16}$ | $2.32 \times 10^{-10}$ | $1.039950 \times 10^{-9}$ | $1.025904 \times 10^{-9}$ | 1.35% | $1.197940 \times 10^{-9}$ | $1.186417 \times 10^{-9}$ | 0.96% |
| 0.219 | $4.27 \times 10^{-16}$ | $4.36 \times 10^{-10}$ | $4.820740 \times 10^{-9}$ | $4.777483 \times 10^{-9}$ | 0.90% | $6.705030 \times 10^{-9}$ | $6.754427 \times 10^{-9}$ | 0.74% |
| 0.222 | $5.09 \times 10^{-16}$ | $2.55 \times 10^{-10}$ | $4.944960 \times 10^{-9}$ | $4.977635 \times 10^{-9}$ | 0.66% | $6.937650 \times 10^{-9}$ | $7.081979 \times 10^{-9}$ | 2.08% |
| 0.214 | $3.01 \times 10^{-16}$ | $1.47 \times 10^{-10}$ | $4.370640 \times 10^{-9}$ | $4.309342 \times 10^{-9}$ | 1.40% | $6.254060 \times 10^{-9}$ | $6.151486 \times 10^{-9}$ | 1.64% |
| 0.209 | $7.12 \times 10^{-16}$ | $2.46 \times 10^{-10}$ | $8.417960 \times 10^{-9}$ | $8.246344 \times 10^{-9}$ | 2.04% | $1.224990 \times 10^{-8}$ | $1.178208 \times 10^{-8}$ | 3.82% |
| 0.206 | $8.52 \times 10^{-16}$ | $1.59 \times 10^{-10}$ | $1.030070 \times 10^{-8}$ | $1.001988 \times 10^{-8}$ | 2.73% | $1.516540 \times 10^{-8}$ | $1.434644 \times 10^{-8}$ | 5.40% |

**Table 17.** Partial data under 28 nm technology (T = 25 °C, $Vbs = 0$).

| | | | $\mu$ (s) | | | $\sigma$ (s) | | |
|---|---|---|---|---|---|---|---|---|
| $Vgs$ (V) | Cl (F) | $\tau$ (s) | $\mu$ (MC) | $\mu$ (Model) | $\mu$ (Error) | $\sigma$ (MC) | $\sigma$ (Model) | $\sigma$ (Error) |
| 0.292 | $9.92 \times 10^{-16}$ | $2.14 \times 10^{-10}$ | $7.646140 \times 10^{-7}$ | $7.651751 \times 10^{-7}$ | 0.07% | $1.805700 \times 10^{-6}$ | $1.796862 \times 10^{-6}$ | 0.49% |
| 0.263 | $7.97 \times 10^{-16}$ | $2.48 \times 10^{-10}$ | $1.357800 \times 10^{-6}$ | $1.365124 \times 10^{-6}$ | 0.54% | $3.193240 \times 10^{-6}$ | $3.205779 \times 10^{-6}$ | 0.39% |
| 0.274 | $3.32 \times 10^{-16}$ | $4.08 \times 10^{-11}$ | $6.337930 \times 10^{-7}$ | $6.328385 \times 10^{-7}$ | 0.15% | $1.490450 \times 10^{-6}$ | $1.486129 \times 10^{-6}$ | 0.29% |
| 0.256 | $4.94 \times 10^{-16}$ | $3.78 \times 10^{-10}$ | $1.209320 \times 10^{-6}$ | $1.213141 \times 10^{-6}$ | 0.32% | $2.839030 \times 10^{-6}$ | $2.848861 \times 10^{-6}$ | 0.35% |
| 0.27 | $4.46 \times 10^{-16}$ | $7.50 \times 10^{-12}$ | $8.085170 \times 10^{-7}$ | $8.090342 \times 10^{-7}$ | 0.06% | $1.901590 \times 10^{-6}$ | $1.899907 \times 10^{-6}$ | 0.09% |
| 0.298 | $7.93 \times 10^{-16}$ | $4.56 \times 10^{-11}$ | $5.643700 \times 10^{-7}$ | $5.629425 \times 10^{-7}$ | 0.25% | $1.333000 \times 10^{-6}$ | $1.321983 \times 10^{-6}$ | 0.83% |
| 0.28 | $8.43 \times 10^{-16}$ | $3.75 \times 10^{-10}$ | $9.226660 \times 10^{-7}$ | $9.252843 \times 10^{-7}$ | 0.28% | $2.174680 \times 10^{-6}$ | $2.172834 \times 10^{-6}$ | 0.08% |
| 0.256 | $1.22 \times 10^{-16}$ | $1.30 \times 10^{-10}$ | $7.046400 \times 10^{-7}$ | $7.069365 \times 10^{-7}$ | 0.33% | $1.640960 \times 10^{-6}$ | $1.660133 \times 10^{-6}$ | 1.17% |
| 0.297 | $1.49 \times 10^{-16}$ | $4.05 \times 10^{-10}$ | $2.673090 \times 10^{-7}$ | $2.651178 \times 10^{-7}$ | 0.82% | $6.295670 \times 10^{-7}$ | $6.224933 \times 10^{-7}$ | 1.12% |
| 0.278 | $5.20 \times 10^{-16}$ | $2.60 \times 10^{-10}$ | $7.199620 \times 10^{-7}$ | $7.197352 \times 10^{-7}$ | 0.03% | $1.695590 \times 10^{-6}$ | $1.690154 \times 10^{-6}$ | 0.32% |

### 4.2.7. Comparison with Other Studies

The propagation delay model proposed in this paper has a high accuracy under the 22 nm FDSOI process, and we compared the other two models [4,18] under the same process. Model [4] provided a very complete model that took temperature into account, but it ignored the influence of DIBL effect, which may make the results inaccurate under different technologies. The model [18] did not simplify Kirchhoff's law, so the Laplace transform and some complex calculations were used in the calculation. Additionally, models [4,18] did not further derive the variance of the delay. The results of the comparison with them are shown in Table 18.

**Table 18.** Propagation delay at different voltages (T = 25 °C, Cl = 0.5 fF, $\tau = 1 \times 10^{-11}$ s, *Vbs* = 0).

| *Vgs* (V) | MC | Model [4] (s) | Error | Model [18] (s) | Error | Model (s) | Error |
|---|---|---|---|---|---|---|---|
| 0.20 | $2.68 \times 10^{-9}$ | $1.12 \times 10^{-9}$ | 48.18% | $2.72 \times 10^{-9}$ | 1.48% | $2.66 \times 10^{-9}$ | 0.85% |
| 0.21 | $2.10 \times 10^{-9}$ | $9.46 \times 10^{-10}$ | 44.08% | $2.15 \times 10^{-9}$ | 2.45% | $2.11 \times 10^{-9}$ | 0.09% |
| 0.22 | $1.65 \times 10^{-9}$ | $7.93 \times 10^{-10}$ | 39.95% | $1.70 \times 10^{-9}$ | 3.23% | $1.67 \times 10^{-9}$ | 0.85% |
| 0.23 | $1.30 \times 10^{-9}$ | $6.62 \times 10^{-10}$ | 35.74% | $1.35 \times 10^{-9}$ | 3.81% | $1.31 \times 10^{-9}$ | 1.41% |
| 0.24 | $1.02 \times 10^{-9}$ | $5.50 \times 10^{-10}$ | 31.48% | $1.06 \times 10^{-9}$ | 4.16% | $1.04 \times 10^{-9}$ | 1.75% |
| 0.25 | $7.99 \times 10^{-10}$ | $4.55 \times 10^{-10}$ | 26.93% | $8.34 \times 10^{-10}$ | 4.40% | $8.15 \times 10^{-10}$ | 1.97% |
| 0.26 | $6.29 \times 10^{-10}$ | $3.75 \times 10^{-10}$ | 22.27% | $6.56 \times 10^{-10}$ | 4.31% | $6.40 \times 10^{-10}$ | 1.88% |
| 0.27 | $4.96 \times 10^{-10}$ | $3.08 \times 10^{-10}$ | 17.77% | $5.15 \times 10^{-10}$ | 3.72% | $5.03 \times 10^{-10}$ | 1.29% |
| 0.28 | $3.93 \times 10^{-10}$ | $2.52 \times 10^{-10}$ | 13.36% | $4.04 \times 10^{-10}$ | 2.69% | $3.94 \times 10^{-10}$ | 0.27% |
| 0.29 | $3.13 \times 10^{-10}$ | $2.05 \times 10^{-10}$ | 9.08% | $3.16 \times 10^{-10}$ | 1.14% | $3.09 \times 10^{-10}$ | 1.25% |
| 0.30 | $2.50 \times 10^{-10}$ | $1.66 \times 10^{-10}$ | 5.01% | $2.48 \times 10^{-10}$ | 0.90% | $2.42 \times 10^{-10}$ | 3.26% |

It is clear that our proposed propagation delay model has a higher accuracy, with the error being only 1.35% in this set of data.

*4.3. Delay Distribution Verification*

In this section, the delay modeling and inverse Gaussian distribution are combined to predict the distribution characteristics of the delay through analytical expressions, eliminating the need for redundant fitting work. Figures 13 and 14 below are the probability density function and cumulative distribution function of the delay distribution at an operating voltage of 0.25 V, 0.27 V, and 0.3 V in Figure 13a–c respectively, from which we can clearly observe the probability density function curve of the model prediction; the fitting curve and the MC simulation results are close to each other, and the error of the CDF is about 2%.

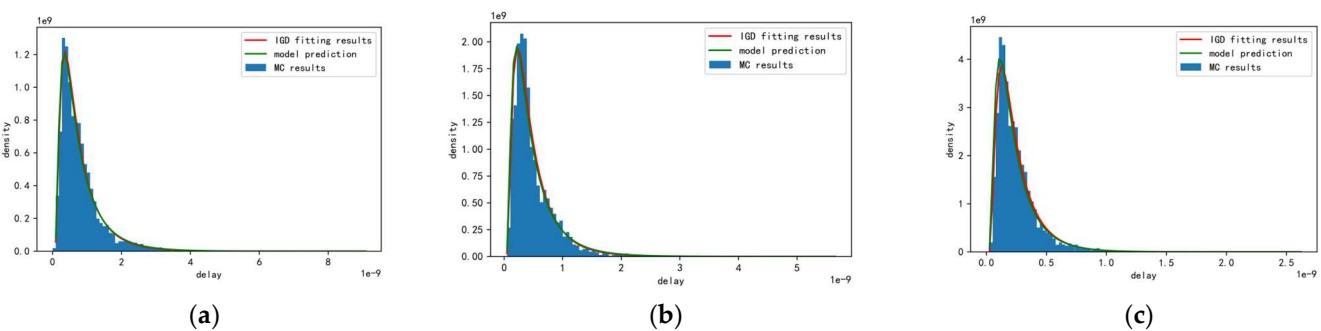

| (a) | (b) | (c) |
|---|---|---|

**Figure 13.** Comparison of PDF curves among MC simulation, IGD fitting, and IGD delay modeling prediction: (**a**) *Vgs* = 0.25 V; (**b**) *Vgs* = 0.27 V; (**c**) *Vgs* = 0.30 V.

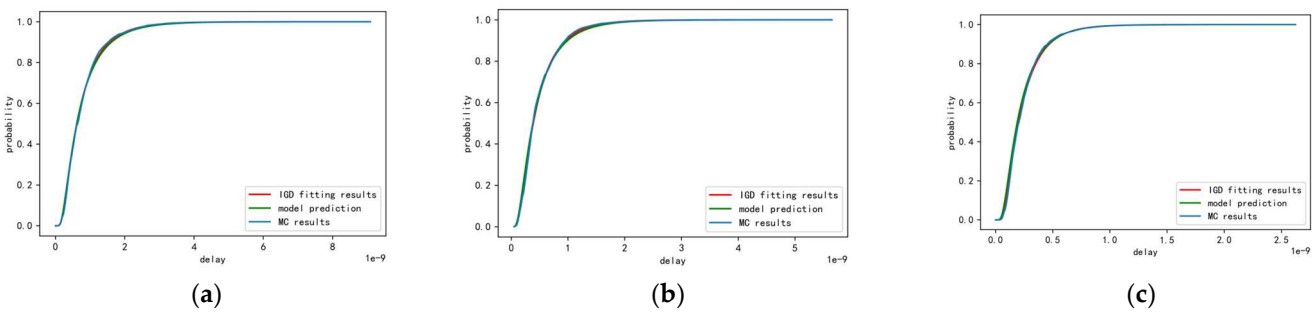

| (a) | (b) | (c) |
|---|---|---|

**Figure 14.** CDF curves comparison among MC simulation, IGD fitting and IGD delay modeling prediction (**a**) *Vgs* = 0.25 V; (**b**) *Vgs* = 0.27 V; (**c**) *Vgs* = 0.30 V.

### 4.4. Speed of the Model

The following Figure 15 shows the curve of the calculation time with the amount of data. For the acquisition of delays, the time is the same for each set of SPICE MC simulation data. As the amount of data increases, the time required increases linearly. The proposed model only needs to perform a DC simulation at the beginning, ($t_{DC} = 1.310$ s), and then an MC simulation ($t_{MC} = 242.624$ s). Then it calculates the corresponding coefficients ($t_{coefficient} = 6.513$ s), and the delay under different inputs can be predicted through the model calculation, with each set of calculation results lasting 0.386 s ($t_{calculation}$). The specific time consumption of the model is calculated as Equation (25), where n represents the amount of delay data.

$$t_{model} = 1 \cdot t_{DC} + 1 \cdot t_{MC} + 1 \cdot t_{coefficient} + n \cdot t_{calculation} = 250.447 \, s + n \cdot 0.386 \, s \quad (25)$$

Although an MC simulation is performed during the preparation process to obtain some coefficients, it is only undertaken once, and the coefficients will not be reacquired when the input changes. Therefore, as the amount of data increases, the time required for model calculation grows relatively slower compared to the SPICE MC simulation.

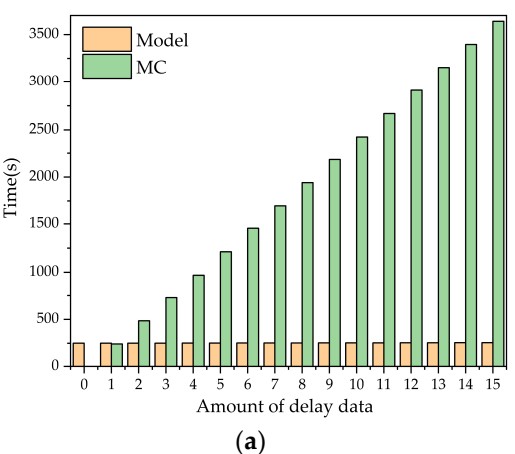
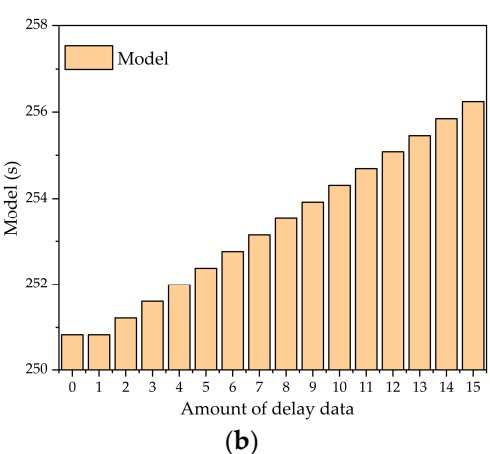

**Figure 15.** Time consumption varies with the amount of data. (**a**) Comparison of the model and SPICE MC simulation; (**b**) enlarged view of the model calculation time in (**a**).

### 5. Conclusions

In this paper, we propose a distribution curve function that accurately predicts the delay of sub-threshold circuits with a high accuracy. The curve of this function is extremely similar to the delay distribution of the circuit. We then derive the key parameters of this function, namely, the mean and variance of the propagation delay, which are derived from the sub-threshold current formula and input–output waveform curves. The results are in good agreement with the SPICE MC simulation from the 22 nm Industrial Design Suite, where the error of the mean and standard deviation for the inverter are 1.5% and 4.3%, respectively. We also verified it with other cell circuits, such as NOR2 gate and NAND2 gate, with all obtaining good results. The derived model parameters are substituted into the inverse Gaussian distribution function, finding that the result is very close to the distribution of Monte Carlo simulations, with the maximum error of the CDF being approximately 2%. In addition, our model is also applicable to other technologies. We have verified the model using 28 nm CMOS and 40 nm CMOS technologies, and the results match the MC simulation results very well. The proposed method only requires performing a DC simulation, to obtain the coefficient of the NMOS and PMOS transistors, and a MC simulation of the circuit under a certain process; no other simulations are required, which greatly reduces the simulation time.

The prediction method can quickly calculate the statistical parameters of delay for cell circuits in the sub-threshold region, which accelerates the statistical characterization and is

very helpful for further evaluating device-level optimization on low-power circuits and architectures. Additionally, based on the study, we will continue to explore the path delay variation of the circuits in the future.

**Author Contributions:** Conceptualization, J.W., Y.W. and X.Z.; methodology, J.W.; software, J.W., Z.L. and L.C.; validation, J.W.; formal analysis, J.W., Y.W. and X.Z.; investigation, J.W.; resources, J.W., Y.W., Z.L., L.C. and X.Z.; data curation, J.W.; writing—original draft preparation, J.W.; writing—review and editing, J.W., Y.W. and X.Z.; supervision, Y.W.; funding acquisition, Y.W., Z.L. and L.C. All authors have read and agreed to the published version of the manuscript.

**Funding:** This research was funded by EDA Center of Institute of Microelectronics, Chinese Academy of Sciences.

**Data Availability Statement:** Not applicable.

**Acknowledgments:** The funding support for our work is greatly appreciated.

**Conflicts of Interest:** The authors declare no conflict of interest.

## Appendix A

Under different sizes, the nominal DC value under different voltages is simulated and compared with the current value calculated by the current formula, and the average error of each size is less than 1%. Figure A1a–d is the current curves of different sizes under different $Vds$ and $Vgs$ voltages. Additionally, Tables A2–A5 show the current data compared with simulation results, as well as errors(settings: $Vgs = 0.3$ V when sweeping $Vds$; $Vds = 0.2$ V when sweeping $Vgs$).

**Table A1.** Coefficients for NMOS in the case of different sizes.

| Size | W (nm) | L (nm) | $Vth$ (V) | $\lambda$ | $I0$ (A) | m (mV/dec) | $V_T$ (V) |
|------|--------|--------|-----------|-----------|----------|------------|-----------|
| Size 1 | 80 | 30 | 0.340 | 0.042 | $6.58 \times 10^{-7}$ | 1.285 | 0.0257 |
| Size 2 | 100 | 40 | 0.347 | 0.030 | $6.98 \times 10^{-7}$ | 1.218 | 0.0257 |
| Size 3 | 135 | 50 | 0.352 | 0.024 | $8.10 \times 10^{-7}$ | 1.185 | 0.0257 |
| Size 4 | 150 | 60 | 0.356 | 0.019 | $7.86 \times 10^{-7}$ | 1.165 | 0.0257 |

**Table A2.** Size 1: Current data compared with simulation results.

| | Current (A) | | | | Current (A) | | |
|---|---|---|---|---|---|---|---|
| Sweep $Vgs$ (V) | DC | Formula | Error | Sweep $Vds$ (V) | DC | Formula | Error |
| 0.10 | $6.648585 \times 10^{-10}$ | $6.793776 \times 10^{-10}$ | 2.18% | 0.10 | $2.195759 \times 10^{-7}$ | $2.195075 \times 10^{-7}$ | 0.03% |
| 0.11 | $9.053548 \times 10^{-10}$ | $9.197036 \times 10^{-10}$ | 1.58% | 0.11 | $2.246291 \times 10^{-7}$ | $2.238606 \times 10^{-7}$ | 0.34% |
| 0.12 | $1.232852 \times 10^{-9}$ | $1.245044 \times 10^{-9}$ | 0.99% | 0.12 | $2.290826 \times 10^{-7}$ | $2.278023 \times 10^{-7}$ | 0.56% |
| 0.13 | $1.678799 \times 10^{-9}$ | $1.685471 \times 10^{-9}$ | 0.40% | 0.13 | $2.330990 \times 10^{-7}$ | $2.314740 \times 10^{-7}$ | 0.70% |
| 0.14 | $2.285979 \times 10^{-9}$ | $2.281697 \times 10^{-9}$ | 0.19% | 0.14 | $2.367952 \times 10^{-7}$ | $2.349730 \times 10^{-7}$ | 0.77% |
| 0.15 | $3.112571 \times 10^{-9}$ | $3.088834 \times 10^{-9}$ | 0.76% | 0.15 | $2.402559 \times 10^{-7}$ | $2.383661 \times 10^{-7}$ | 0.79% |
| 0.16 | $4.237630 \times 10^{-9}$ | $4.181492 \times 10^{-9}$ | 1.32% | 0.16 | $2.435429 \times 10^{-7}$ | $2.416996 \times 10^{-7}$ | 0.76% |
| 0.17 | $5.768473 \times 10^{-9}$ | $5.660671 \times 10^{-9}$ | 1.87% | 0.17 | $2.467016 \times 10^{-7}$ | $2.450051 \times 10^{-7}$ | 0.69% |
| 0.18 | $7.850596 \times 10^{-9}$ | $7.663102 \times 10^{-9}$ | 2.39% | 0.18 | $2.497656 \times 10^{-7}$ | $2.483048 \times 10^{-7}$ | 0.58% |
| 0.19 | $1.068089 \times 10^{-8}$ | $1.037388 \times 10^{-8}$ | 2.87% | 0.19 | $2.527602 \times 10^{-7}$ | $2.516139 \times 10^{-7}$ | 0.45% |
| 0.20 | $1.452518 \times 10^{-8}$ | $1.404358 \times 10^{-8}$ | 3.32% | 0.20 | $2.557046 \times 10^{-7}$ | $2.549430 \times 10^{-7}$ | 0.30% |
| 0.21 | $1.974115 \times 10^{-8}$ | $1.901142 \times 10^{-8}$ | 3.70% | 0.21 | $2.586135 \times 10^{-7}$ | $2.582996 \times 10^{-7}$ | 0.12% |
| 0.22 | $2.680801 \times 10^{-8}$ | $2.573661 \times 10^{-8}$ | 4.00% | 0.22 | $2.614981 \times 10^{-7}$ | $2.616892 \times 10^{-7}$ | 0.07% |
| 0.23 | $3.636410 \times 10^{-8}$ | $3.484079 \times 10^{-8}$ | 4.19% | 0.23 | $2.643674 \times 10^{-7}$ | $2.651154 \times 10^{-7}$ | 0.28% |
| 0.24 | $4.925328 \times 10^{-8}$ | $4.716552 \times 10^{-8}$ | 4.24% | 0.24 | $2.672284 \times 10^{-7}$ | $2.685812 \times 10^{-7}$ | 0.51% |
| 0.25 | $6.658004 \times 10^{-8}$ | $6.385006 \times 10^{-8}$ | 4.10% | 0.25 | $2.700866 \times 10^{-7}$ | $2.720886 \times 10^{-7}$ | 0.74% |
| 0.26 | $8.977187 \times 10^{-8}$ | $8.643666 \times 10^{-8}$ | 3.72% | 0.26 | $2.729465 \times 10^{-7}$ | $2.756392 \times 10^{-7}$ | 0.99% |
| 0.27 | $1.206443 \times 10^{-7}$ | $1.170131 \times 10^{-7}$ | 3.01% | 0.27 | $2.758119 \times 10^{-7}$ | $2.792345 \times 10^{-7}$ | 1.24% |
| 0.28 | $1.614615 \times 10^{-7}$ | $1.584059 \times 10^{-7}$ | 1.89% | 0.28 | $2.786857 \times 10^{-7}$ | $2.828756 \times 10^{-7}$ | 1.50% |
| 0.29 | $2.149806 \times 10^{-7}$ | $2.144411 \times 10^{-7}$ | 0.25% | 0.29 | $2.815705 \times 10^{-7}$ | $2.865632 \times 10^{-7}$ | 1.77% |
| 0.30 | $2.844683 \times 10^{-7}$ | $2.902984 \times 10^{-7}$ | 2.05% | 0.30 | $2.844683 \times 10^{-7}$ | $2.902984 \times 10^{-7}$ | 2.05% |

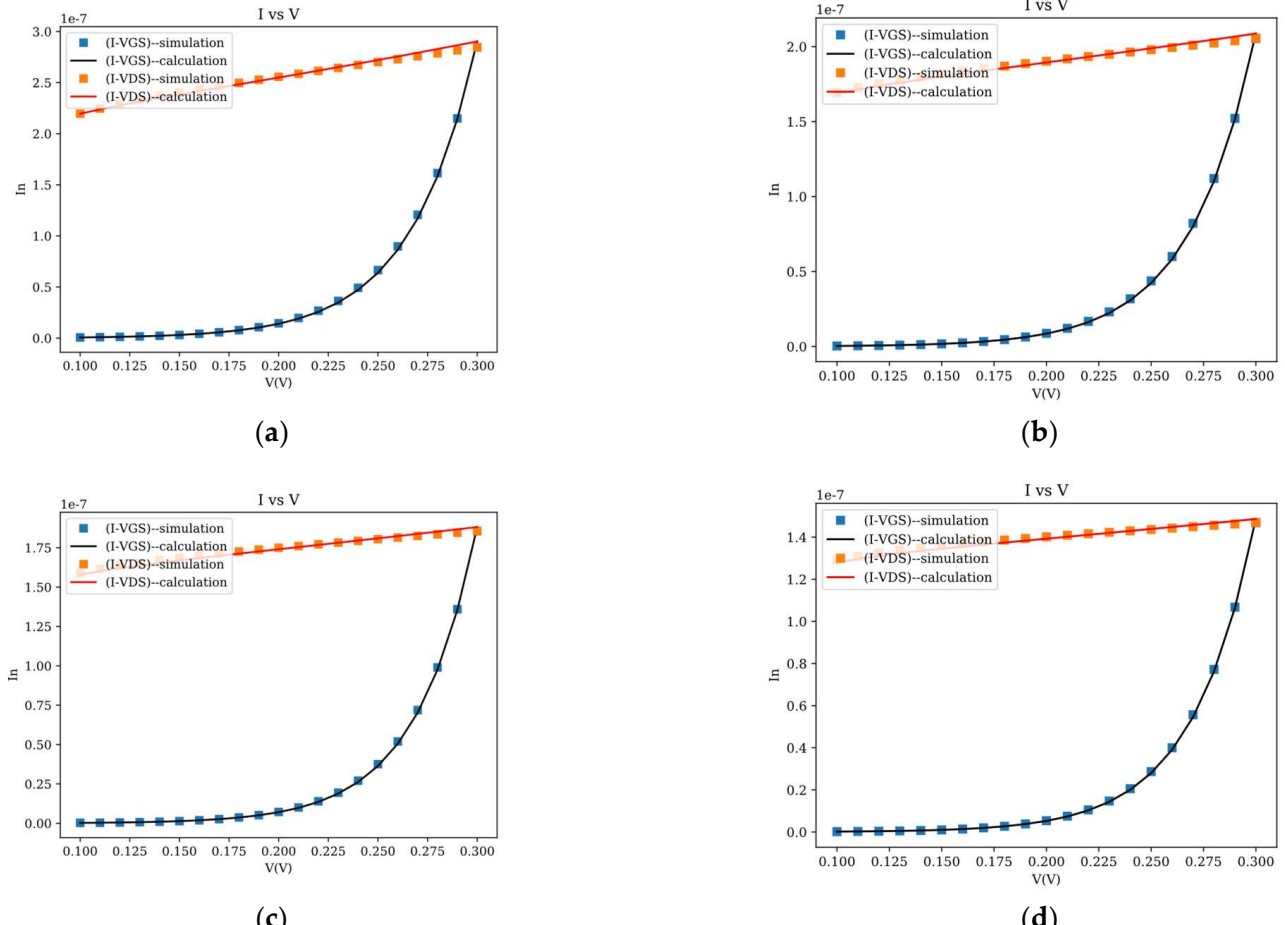

(**a**)  (**b**)

(**c**)  (**d**)

**Figure A1.** Current changes of transistors with voltage, where the symbols represent the DC simulation results and the lines represents the current formula calculation results. (**a**) Size 1; (**b**) size 2; (**c**) Size 3; (**d**) Size 4.

**Table A3.** Size 2: Current data compared with simulation results.

| | Current (A) | | | | Current (A) | | |
|---|---|---|---|---|---|---|---|
| **Sweep Vgs** | **DC** | **Formula** | **Error** | **Sweep Vds** | **DC** | **Formula** | **Error** |
| 0.10 | $3.447808 \times 10^{-10}$ | $3.505929 \times 10^{-10}$ | 1.69% | 0.10 | $1.693521 \times 10^{-7}$ | $1.685674 \times 10^{-7}$ | 0.46% |
| 0.11 | $4.766647 \times 10^{-10}$ | $4.825496 \times 10^{-10}$ | 1.23% | 0.11 | $1.724931 \times 10^{-7}$ | $1.713454 \times 10^{-7}$ | 0.67% |
| 0.12 | $6.590156 \times 10^{-10}$ | $6.641723 \times 10^{-10}$ | 0.78% | 0.12 | $1.751957 \times 10^{-7}$ | $1.737894 \times 10^{-7}$ | 0.80% |
| 0.13 | $9.111418 \times 10^{-10}$ | $9.141543 \times 10^{-10}$ | 0.33% | 0.13 | $1.775811 \times 10^{-7}$ | $1.760102 \times 10^{-7}$ | 0.88% |
| 0.14 | $1.259729 \times 10^{-9}$ | $1.258225 \times 10^{-9}$ | 0.12% | 0.14 | $1.797351 \times 10^{-7}$ | $1.780836 \times 10^{-7}$ | 0.92% |
| 0.15 | $1.741646 \times 10^{-9}$ | $1.731797 \times 10^{-9}$ | 0.57% | 0.15 | $1.817190 \times 10^{-7}$ | $1.800616 \times 10^{-7}$ | 0.91% |
| 0.16 | $2.407814 \times 10^{-9}$ | $2.383613 \times 10^{-9}$ | 1.01% | 0.16 | $1.835767 \times 10^{-7}$ | $1.819797 \times 10^{-7}$ | 0.87% |
| 0.17 | $3.328513 \times 10^{-9}$ | $3.280761 \times 10^{-9}$ | 1.43% | 0.17 | $1.853401 \times 10^{-7}$ | $1.838624 \times 10^{-7}$ | 0.80% |
| 0.18 | $4.600657 \times 10^{-9}$ | $4.515578 \times 10^{-9}$ | 1.85% | 0.18 | $1.870328 \times 10^{-7}$ | $1.857262 \times 10^{-7}$ | 0.70% |
| 0.19 | $6.357736 \times 10^{-9}$ | $6.215158 \times 10^{-9}$ | 2.24% | 0.19 | $1.886720 \times 10^{-7}$ | $1.875829 \times 10^{-7}$ | 0.58% |
| 0.20 | $8.783299 \times 10^{-9}$ | $8.554427 \times 10^{-9}$ | 2.61% | 0.20 | $1.902708 \times 10^{-7}$ | $1.894402 \times 10^{-7}$ | 0.44% |
| 0.21 | $1.212916 \times 10^{-8}$ | $1.177415 \times 10^{-8}$ | 2.93% | 0.21 | $1.918392 \times 10^{-7}$ | $1.913037 \times 10^{-7}$ | 0.28% |
| 0.22 | $1.673970 \times 10^{-8}$ | $1.620573 \times 10^{-8}$ | 3.19% | 0.22 | $1.933846 \times 10^{-7}$ | $1.931772 \times 10^{-7}$ | 0.11% |
| 0.23 | $2.308387 \times 10^{-8}$ | $2.230526 \times 10^{-8}$ | 3.37% | 0.23 | $1.949129 \times 10^{-7}$ | $1.950633 \times 10^{-7}$ | 0.08% |
| 0.24 | $3.179661 \times 10^{-8}$ | $3.070054 \times 10^{-8}$ | 3.45% | 0.24 | $1.964288 \times 10^{-7}$ | $1.969639 \times 10^{-7}$ | 0.27% |
| 0.25 | $4.373098 \times 10^{-8}$ | $4.225565 \times 10^{-8}$ | 3.37% | 0.25 | $1.979359 \times 10^{-7}$ | $1.988803 \times 10^{-7}$ | 0.48% |
| 0.26 | $6.002143 \times 10^{-8}$ | $5.815989 \times 10^{-8}$ | 3.10% | 0.26 | $1.994370 \times 10^{-7}$ | $2.008136 \times 10^{-7}$ | 0.69% |
| 0.27 | $8.215675 \times 10^{-8}$ | $8.005019 \times 10^{-8}$ | 2.56% | 0.27 | $2.009346 \times 10^{-7}$ | $2.027644 \times 10^{-7}$ | 0.91% |
| 0.28 | $1.120578 \times 10^{-7}$ | $1.101796 \times 10^{-7}$ | 1.68% | 0.28 | $2.024305 \times 10^{-7}$ | $2.047333 \times 10^{-7}$ | 1.14% |
| 0.29 | $1.521509 \times 10^{-7}$ | $1.516491 \times 10^{-7}$ | 0.33% | 0.29 | $2.039263 \times 10^{-7}$ | $2.067207 \times 10^{-7}$ | 1.37% |
| 0.30 | $2.054233 \times 10^{-7}$ | $2.087270 \times 10^{-7}$ | 1.61% | 0.30 | $2.054233 \times 10^{-7}$ | $2.087270 \times 10^{-7}$ | 1.61% |

**Table A4.** Size 3: Current data compared with simulation results.

| | Current (A) | | | | Current (A) | | |
|---|---|---|---|---|---|---|---|
| Sweep $Vgs$ | DC | Formula | Error | Sweep $Vds$ | DC | Formula | Error |
| 0.10 | $2.600795 \times 10^{-10}$ | $2.637755 \times 10^{-10}$ | 1.42% | 0.10 | $1.589399 \times 10^{-7}$ | $1.578282 \times 10^{-7}$ | 0.70% |
| 0.11 | $3.625443 \times 10^{-10}$ | $3.663475 \times 10^{-10}$ | 1.05% | 0.11 | $1.614898 \times 10^{-7}$ | $1.601232 \times 10^{-7}$ | 0.85% |
| 0.12 | $5.053995 \times 10^{-10}$ | $5.088057 \times 10^{-10}$ | 0.67% | 0.12 | $1.636417 \times 10^{-7}$ | $1.620975 \times 10^{-7}$ | 0.94% |
| 0.13 | $7.045667 \times 10^{-10}$ | $7.066603 \times 10^{-10}$ | 0.30% | 0.13 | $1.655069 \times 10^{-7}$ | $1.638558 \times 10^{-7}$ | 1.00% |
| 0.14 | $9.822389 \times 10^{-10}$ | $9.814528 \times 10^{-10}$ | 0.08% | 0.14 | $1.671636 \times 10^{-7}$ | $1.654699 \times 10^{-7}$ | 1.01% |
| 0.15 | $1.369344 \times 10^{-9}$ | $1.363101 \times 10^{-9}$ | 0.46% | 0.15 | $1.686669 \times 10^{-7}$ | $1.669887 \times 10^{-7}$ | 0.99% |
| 0.16 | $1.908969 \times 10^{-9}$ | $1.893158 \times 10^{-9}$ | 0.83% | 0.16 | $1.700564 \times 10^{-7}$ | $1.684457 \times 10^{-7}$ | 0.95% |
| 0.17 | $2.661110 \times 10^{-9}$ | $2.629334 \times 10^{-9}$ | 1.19% | 0.17 | $1.713604 \times 10^{-7}$ | $1.698638 \times 10^{-7}$ | 0.87% |
| 0.18 | $3.709259 \times 10^{-9}$ | $3.651778 \times 10^{-9}$ | 1.55% | 0.18 | $1.725996 \times 10^{-7}$ | $1.712585 \times 10^{-7}$ | 0.78% |
| 0.19 | $5.169487 \times 10^{-9}$ | $5.071812 \times 10^{-9}$ | 1.89% | 0.19 | $1.737895 \times 10^{-7}$ | $1.726406 \times 10^{-7}$ | 0.66% |
| 0.20 | $7.202954 \times 10^{-9}$ | $7.044042 \times 10^{-9}$ | 2.21% | 0.20 | $1.749414 \times 10^{-7}$ | $1.740175 \times 10^{-7}$ | 0.53% |
| 0.21 | $1.003300 \times 10^{-8}$ | $9.783194 \times 10^{-9}$ | 2.49% | 0.21 | $1.760639 \times 10^{-7}$ | $1.753942 \times 10^{-7}$ | 0.38% |
| 0.22 | $1.396839 \times 10^{-8}$ | $1.358750 \times 10^{-8}$ | 2.73% | 0.22 | $1.771637 \times 10^{-7}$ | $1.767741 \times 10^{-7}$ | 0.22% |
| 0.23 | $1.943441 \times 10^{-8}$ | $1.887114 \times 10^{-8}$ | 2.90% | 0.23 | $1.782457 \times 10^{-7}$ | $1.781596 \times 10^{-7}$ | 0.05% |
| 0.24 | $2.701415 \times 10^{-8}$ | $2.620939 \times 10^{-8}$ | 2.98% | 0.24 | $1.793139 \times 10^{-7}$ | $1.795525 \times 10^{-7}$ | 0.13% |
| 0.25 | $3.750178 \times 10^{-8}$ | $3.640120 \times 10^{-8}$ | 2.93% | 0.25 | $1.803715 \times 10^{-7}$ | $1.809538 \times 10^{-7}$ | 0.32% |
| 0.26 | $5.196958 \times 10^{-8}$ | $5.055620 \times 10^{-8}$ | 2.72% | 0.26 | $1.814209 \times 10^{-7}$ | $1.823643 \times 10^{-7}$ | 0.52% |
| 0.27 | $7.184863 \times 10^{-8}$ | $7.021553 \times 10^{-8}$ | 2.27% | 0.27 | $1.824642 \times 10^{-7}$ | $1.837847 \times 10^{-7}$ | 0.72% |
| 0.28 | $9.902015 \times 10^{-8}$ | $9.751960 \times 10^{-8}$ | 1.52% | 0.28 | $1.835028 \times 10^{-7}$ | $1.852154 \times 10^{-7}$ | 0.93% |
| 0.29 | $1.359099 \times 10^{-7}$ | $1.354412 \times 10^{-7}$ | 0.34% | 0.29 | $1.845383 \times 10^{-7}$ | $1.866568 \times 10^{-7}$ | 1.15% |
| 0.30 | $1.855716 \times 10^{-7}$ | $1.881089 \times 10^{-7}$ | 1.37% | 0.30 | $1.855716 \times 10^{-7}$ | $1.881089 \times 10^{-7}$ | 1.37% |

**Table A5.** Size 4: Current data compared with simulation results.

| | Current (A) | | | | Current (A) | | |
|---|---|---|---|---|---|---|---|
| Sweep $Vgs$ | DC | Formula | Error | Sweep $Vds$ | DC | Formula | Error |
| 0.10 | $1.840426 \times 10^{-10}$ | $1.862541 \times 10^{-10}$ | 1.20% | 0.10 | $1.291076 \times 10^{-7}$ | $1.279476 \times 10^{-7}$ | 0.90% |
| 0.11 | $2.578295 \times 10^{-10}$ | $2.601388 \times 10^{-10}$ | 0.90% | 0.11 | $1.309655 \times 10^{-7}$ | $1.296413 \times 10^{-7}$ | 1.01% |
| 0.12 | $3.612193 \times 10^{-10}$ | $3.633325 \times 10^{-10}$ | 0.59% | 0.12 | $1.325076 \times 10^{-7}$ | $1.310711 \times 10^{-7}$ | 1.08% |
| 0.13 | $5.060898 \times 10^{-10}$ | $5.074618 \times 10^{-10}$ | 0.27% | 0.13 | $1.338230 \times 10^{-7}$ | $1.323226 \times 10^{-7}$ | 1.12% |
| 0.14 | $7.090832 \times 10^{-10}$ | $7.087654 \times 10^{-10}$ | 0.04% | 0.14 | $1.349737 \times 10^{-7}$ | $1.334544 \times 10^{-7}$ | 1.13% |
| 0.15 | $9.935122 \times 10^{-10}$ | $9.899235 \times 10^{-10}$ | 0.36% | 0.15 | $1.360035 \times 10^{-7}$ | $1.345063 \times 10^{-7}$ | 1.10% |
| 0.16 | $1.392026 \times 10^{-9}$ | $1.382613 \times 10^{-9}$ | 0.68% | 0.16 | $1.369433 \times 10^{-7}$ | $1.355055 \times 10^{-7}$ | 1.05% |
| 0.17 | $1.950334 \times 10^{-9}$ | $1.931078 \times 10^{-9}$ | 0.99% | 0.17 | $1.378154 \times 10^{-7}$ | $1.364707 \times 10^{-7}$ | 0.98% |
| 0.18 | $2.732393 \times 10^{-9}$ | $2.697113 \times 10^{-9}$ | 1.29% | 0.18 | $1.386360 \times 10^{-7}$ | $1.374144 \times 10^{-7}$ | 0.88% |
| 0.19 | $3.827632 \times 10^{-9}$ | $3.767022 \times 10^{-9}$ | 1.58% | 0.19 | $1.394172 \times 10^{-7}$ | $1.383454 \times 10^{-7}$ | 0.77% |
| 0.20 | $5.360954 \times 10^{-9}$ | $5.261352 \times 10^{-9}$ | 1.86% | 0.20 | $1.401677 \times 10^{-7}$ | $1.392696 \times 10^{-7}$ | 0.64% |
| 0.21 | $7.506557 \times 10^{-9}$ | $7.348462 \times 10^{-9}$ | 2.11% | 0.21 | $1.408941 \times 10^{-7}$ | $1.401910 \times 10^{-7}$ | 0.50% |
| 0.22 | $1.050688 \times 10^{-8}$ | $1.026350 \times 10^{-8}$ | 2.32% | 0.22 | $1.416017 \times 10^{-7}$ | $1.411124 \times 10^{-7}$ | 0.35% |
| 0.23 | $1.469834 \times 10^{-8}$ | $1.433949 \times 10^{-8}$ | 2.47% | 0.23 | $1.422943 \times 10^{-7}$ | $1.420357 \times 10^{-7}$ | 0.18% |
| 0.24 | $2.054593 \times 10^{-8}$ | $2.002137 \times 10^{-8}$ | 2.55% | 0.24 | $1.429750 \times 10^{-7}$ | $1.429622 \times 10^{-7}$ | 0.01% |
| 0.25 | $2.868881 \times 10^{-8}$ | $2.796360 \times 10^{-8}$ | 2.53% | 0.25 | $1.436461 \times 10^{-7}$ | $1.438928 \times 10^{-7}$ | 0.17% |
| 0.26 | $3.999898 \times 10^{-8}$ | $3.905640 \times 10^{-8}$ | 2.36% | 0.26 | $1.443096 \times 10^{-7}$ | $1.448281 \times 10^{-7}$ | 0.36% |
| 0.27 | $5.565421 \times 10^{-8}$ | $5.454957 \times 10^{-8}$ | 1.98% | 0.27 | $1.449670 \times 10^{-7}$ | $1.457686 \times 10^{-7}$ | 0.55% |
| 0.28 | $7.722428 \times 10^{-8}$ | $7.618868 \times 10^{-8}$ | 1.34% | 0.28 | $1.456195 \times 10^{-7}$ | $1.467146 \times 10^{-7}$ | 0.75% |
| 0.29 | $1.067656 \times 10^{-7}$ | $1.064118 \times 10^{-7}$ | 0.33% | 0.29 | $1.462681 \times 10^{-7}$ | $1.476663 \times 10^{-7}$ | 0.96% |
| 0.30 | $1.469137 \times 10^{-7}$ | $1.486239 \times 10^{-7}$ | 1.16% | 0.30 | $1.469137 \times 10^{-7}$ | $1.486239 \times 10^{-7}$ | 1.16% |

## Appendix B

For Equation (A1) to (A2):

$$\frac{dVout(t)}{e^{\frac{\lambda_n Vout(t)}{m_n V_T}}} = \frac{-I_{0n} \cdot e^{\frac{-Vthb - Vthn}{m_n V_T}}}{C_{tot}} \cdot e^{\frac{Vdd}{\tau}t}{m_n V_T} \, dt \tag{A1}$$

$$Vout(t) = \frac{-m_n V_T}{\lambda_n} \cdot ln\left( \frac{I_{0n} \cdot e^{\frac{-Vthb - Vthn}{m_n V_T}} \lambda_n \tau}{Vdd \cdot C_{tot}} \left( e^{\frac{Vdd}{m_n V_T \tau}t} - 1 \right) + e^{-\frac{\lambda_n Vdd}{m_n V_T}} \right) \tag{A2}$$

Details:
Integrate the left side of Equation (A1):

$$\int \frac{dVout(t)}{e^{\frac{\lambda_n Vout(t)}{m_n V_T}}} = \frac{-m_n V_T}{\lambda_n} e^{\frac{-\lambda_n Vout(t)}{m_n V_T}} \tag{A3}$$

Integrate the right side of Equation (A1):

$$\int \frac{-I_{0n} \cdot e^{\frac{-Vthb-Vthn}{m_n V_T}}}{C_{tot}} \cdot e^{\frac{Vdd}{\tau} t}{m_n V_T} \, dt = \frac{-I_{0n} \cdot e^{\frac{-Vthb-Vthn}{m_n V_T}} m_n V_T \tau}{C_{tot} Vdd} e^{\frac{Vdd}{\tau} t}{m_n V_T} \tag{A4}$$

Then, we can obtain

$$\frac{-m_n V_T}{\lambda_n} e^{\frac{-\lambda_n Vout(t)}{m_n V_T}} = \frac{-I_{0n} \cdot e^{\frac{-Vthb-Vthn}{m_n V_T}} m_n V_T \tau}{C_{tot} Vdd} e^{\frac{Vdd}{\tau} t}{m_n V_T} + C1 \tag{A5}$$

where $C1$ is a constant. Then, we substitute the initial condition $Vout(t) = Vdd$ when $t = 0$, and we can obtain $C1$:

$$C1 = \frac{I_{0n} \cdot e^{\frac{-Vthb-Vthn}{m_n V_T}} m_n V_T \tau}{C_{tot} Vdd} - \frac{m_n V_T}{\lambda_n} e^{\frac{-\lambda_n Vdd}{m_n V_T}} \tag{A6}$$

Then, substitute $C1$ and we can obtain Equation (A2).
For Equation (A7): $t > \tau$

$$Vout(t) = \frac{-m_n V_T}{\lambda_n} \cdot ln \left( \frac{I_{0n} \cdot e^{\frac{-Vthb-Vthn}{m_n V_T}} \lambda_n}{C_{tot} m_n V_T} e^{\frac{Vdd}{m_n V_T}} (t - \tau) + e^{-\frac{\lambda_n Vout(\tau)}{m_n V_T}} \right) \tag{A7}$$

Details:
In this case, $dVin(t) = 0$, $Vin = Vdd$; substitute them into Equation (8) and phase shift, then we can obtain

$$\frac{dVout(t)}{e^{\frac{\lambda_n Vout(t)}{m_n V_T}}} = \frac{-I_{0n} \cdot e^{\frac{-Vthb-Vthn}{m_n V_T}}}{C_{tot}} e^{\frac{Vdd}{m_n V_T}} \, dt \tag{A8}$$

Integrate the left side of the equation:

$$\int \frac{dVout(t)}{e^{\frac{\lambda_n Vout(t)}{m_n V_T}}} = \frac{-m_n V_T}{\lambda_n} e^{\frac{-\lambda_n Vout(t)}{m_n V_T}} \tag{A9}$$

$$\int \frac{-I_{0n} \cdot e^{\frac{-Vthb-Vthn}{m_n V_T}}}{C_{tot}} e^{\frac{Vdd}{m_n V_T}} \, dt = \frac{-I_{0n} \cdot e^{\frac{-Vthb-Vthn}{m_n V_T}}}{C_{tot}} e^{\frac{Vdd}{m_n V_T}} \cdot t \tag{A10}$$

Integrate the right side of the equation:
Then, we can obtain

$$\frac{-m_n V_T}{\lambda_n} e^{\frac{-\lambda_n Vout(t)}{m_n V_T}} = \frac{-I_{0n} \cdot e^{\frac{-Vthb-Vthn}{m_n V_T}}}{C_{tot}} e^{\frac{Vdd}{m_n V_T}} \cdot t + C2 \tag{A11}$$

where $C2$ is a constant, then substitute the initial condition $t = \tau$, $Vout(t) = Vout(\tau)$, and we can get the value of $C2$:

$$C2 = \frac{I_{0n} \cdot e^{\frac{-Vthb-Vthn}{m_n V_T}}}{C_{tot}} e^{\frac{Vdd \cdot \tau}{m_n V_T}} - \frac{m_n V_T}{\lambda_n} e^{\frac{-\lambda_n Vout(\tau)}{m_n V_T}} \tag{A12}$$

Then, substitute $C2$ and we can obtain the expression (A7).

For Equation (A13) to (A14):

$$Td = k0 \cdot \frac{C_{tot} m_n V_T}{I_{0n} \cdot e^{\frac{-Vthb - Vthn}{m_n V_T}} \lambda_n} e^{\frac{-Vdd}{m_n V_T}} \left( e^{-\frac{\lambda_n Vdd}{2m_n V_T}} - e^{-\frac{\lambda_n Vout(\tau)}{m_n V_T}} \right) + \frac{\tau}{2} \tag{A13}$$

$$Td = k0 \cdot \frac{C_{tot} m_n V_T}{I_{0n} \cdot e^{\frac{-Vthb - Vthn}{m_n V_T}} \lambda_n} e^{\frac{-Vdd}{m_n V_T}} \left( e^{-\frac{\lambda_n Vdd}{2m_n V_T}} - e^{-\frac{\lambda_n Vdd}{m_n V_T}} \right) + \tau \cdot [\frac{1}{2} - k0 \cdot \frac{m_n V_T}{Vdd} \cdot \left( 1 - e^{\frac{-Vdd}{m_n V_T}} \right)] \tag{A14}$$

Substitute the equation of $Vout(\tau)$ to $e^{-\frac{\lambda_n Vout(\tau)}{m_n V_T}}$:

$$e^{-\frac{\lambda_n Vout(\tau)}{m_n V_T}} = \left( \frac{I_{0n} \cdot e^{\frac{-Vthb - Vthn}{m_n V_T}} \lambda_n \tau}{Vdd C_{tot}} \left( e^{\frac{Vdd}{m_n V_T}} - 1 \right) + e^{-\frac{\lambda_n Vdd}{m_n V_T}} \right) \tag{A15}$$

Then, substitute (A15) to Equation (A13):

$$
\begin{aligned}
Td = t - \frac{\tau}{2} &= k0 \cdot \frac{C_{tot} m_n V_T}{I_{0n} \cdot e^{\frac{-Vthb - Vthn}{m_n V_T}} \lambda_n} e^{\frac{-Vdd}{m_n V_T}} \left( e^{-\frac{\lambda_n Vdd}{2m_n V_T}} - e^{-\frac{\lambda_n Vout(\tau)}{m_n V_T}} \right) + \frac{\tau}{2} \\
&= k0 \cdot \frac{C_{tot} m_n V_T}{I_{0n} \cdot e^{\frac{-Vthb - Vthn}{m_n V_T}} \lambda_n} e^{\frac{-Vdd}{m_n V_T}} \left( e^{-\frac{\lambda_n Vdd}{2m_n V_T}} - \frac{I_{0n} \cdot e^{\frac{-Vthb - Vthn}{m_n V_T}} \lambda_n \tau}{Vdd C_{tot}} \left( e^{\frac{Vdd}{m V_T}} - 1 \right) - e^{-\frac{\lambda_n Vdd}{m_n V_T}} \right) + \frac{\tau}{2} \\
&= k0 \cdot \frac{C_{tot} m_n V_T}{I_{0n} \cdot e^{\frac{-Vthb - Vthn}{m_n V_T}} \lambda_n} e^{\frac{-Vdd}{m_n V_T}} \left( e^{-\frac{\lambda_n Vdd}{2m_n V_T}} - e^{-\frac{\lambda_n Vdd}{m_n V_T}} \right) - k0 \cdot \frac{m_n V_T}{Vdd} \tau \cdot e^{\frac{-Vdd}{m_n V_T}} \left( e^{\frac{Vdd}{m V_T}} - 1 \right) + \frac{\tau}{2} \\
&= k0 \cdot \frac{c_{tot} m_n V_T}{I_{0n} \cdot e^{\frac{-Vthb - Vthn}{m_n V_T}} \lambda_n} e^{\frac{-Vdd}{m_n V_T}} \left( e^{-\frac{\lambda_n Vdd}{2m_n V_T}} - e^{-\frac{\lambda_n Vdd}{m_n V_T}} \right) + \tau \cdot \left[ \frac{1}{2} - k0 \cdot \frac{m_n V_T}{Vdd} \cdot \left( 1 - e^{\frac{-Vdd}{m_n V_T}} \right) \right]
\end{aligned}
\tag{A16}
$$

Then, we can obtain Equation (A14).
For Equation (A17) to (A18):

$$Td = t1 - \frac{\tau}{2} = k0 \frac{m_n V_T \tau}{Vdd} ln[\frac{Vdd \cdot C_{tot}}{I_{0n} \cdot e^{\frac{-Vthb - Vthn}{m_n V_T}} \lambda_n \tau} (e^{\frac{-\lambda_n Vdd}{2m_n V_T}} - e^{-\frac{\lambda_n Vdd}{m_n V_T}}) + 1] - \frac{\tau}{2} \tag{A17}$$

$$\sigma^2(Td) = \left( k0 \frac{\tau}{Vdd} \right)^2 \sigma^2(Vthb + Vthn) \tag{A18}$$

Details:
In order separate the threshold voltage, we neglect "1" in (A17), then we can obtain

$$Td \sim k0 \frac{m_n V_T \tau}{Vdd} ln(\frac{Vdd C_{tot}}{I_{0n} \cdot e^{\frac{-Vthb - Vthn}{m_n V_T}} \lambda_n \tau}) \sim k0 \frac{m_n V_T \tau}{Vdd} ln(\cdot e^{\frac{Vthb + Vthn}{m_n V_T}}) \sim k0 \frac{m_n V_T \tau}{Vdd} \cdot \frac{Vthb + Vthn}{m_n V_T} \sim k0 \frac{\tau}{Vdd} (Vthb + Vthn) \tag{A19}$$

Then, we can obtain

$$\sigma^2(Td) = (k0 \frac{\tau}{Vdd})^2 \sigma^2(Vthb + Vthn) \tag{A20}$$

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
