# Peer review of "Analytical Delay Modeling for a Sub-Threshold Cell Circuit with the Inverse Gaussian Distribution Function"

_electronics, doi:10.3390/electronics12061387_

Round 1

Reviewer 1 Report

The authors of this work have analyzed the delay of subthreshold cells by which a better estimation of delay will be available after designing the cell. The materials are worth to be considered, however, the following concerns should be addressed in detail.

1-     The introduction should be extended and included more works related to subthreshold circuit and parameter variations. Regarding the title of this work, it is worth considering the following references in which MOS current mismatch and subthreshold region have been analyzed.

M. Akbari, O. Hashemipour and F. Moradi, "Input Offset Estimation of CMOS Integrated Circuits in Weak Inversion," in IEEE Transactions on Very Large-Scale Integration (VLSI) Systems, vol. 26, no. 9, pp. 1812-1816, Sept. 2018.

M. Alioto, “Understanding DC behavior of subthreshold CMOS logic through closed-form analysis,” IEEE Trans. Circuits Syst. I, Reg. Papers, vol. 57, no. 7, pp. 1597–1607, Jul. 2012.

2-     Equations (7) and (8), how much are accurate for short channel transistors? You are using 22-nm technology and the equations should be accurate.

3-     Why have not you used EKV model by which all operating regions can be covered such that your analyses will be valid in all regions.

4-     Some details of the mathematical calculations which are not visible can be added as appendix.

5-     How and where have you found the values of some parameters to put in the equations and compare them with simulations results? Figs. 7 and 8. How did you extract the parameters from technology node? Some of them may be variable under different biasing points and transient responses.

6-     How much the calculations will be matched with simulations for different transistor sizing? How did you find the optimum sizing by which achieving a good matching with simulations?

7-     How much your analysis will be accurate in different technology nodes? Have you simulated based on other technologies too?

8-     What kinds of cells can be designed by your method? Can you provide a list of them with your reasons?

Author Response

Dear Reviewer:

We gratefully appreciate your valuable suggestions. Our responses to the comments are shown below, we will respond to your comments and suggestions point by point. The explanation part of the reply is in blue, and the revised manuscript content is in red. And revisions to the manuscript are marked up using the “Track Changes” function of MS Word version, such that any changes can be easily viewed by the editors and reviewers.

If there are any problems, please let us know. Thank you again.

Best regards,

Yours sincerely.

Prof. Yuping Wu(Corresponding author)

University of Chinese Academy of Sciences, Beijing 100049, China;

Institute of Microelectronics of Chinese Academy of Sciences, N0.03 Beitucheng West Road, Chaoyang District, Beijing, PR China. E-Mail: [email protected]

Phone: +86-010-82995693

Reviewer 2 Report

While the work is related to the most basic non linear circuits - the cmos inverter - it is really interesting and well presented. This topic is not a topic where state of the art of circuit design can be addressed but the basics vs non linear operation of MOS transistors.

I like the analysis and the full structure of the work - the simulation work presented is advanced and extremely detailed. I do not see however any comparison with other related works oe application where this work can be applied. This could be on simulation and modeling.

I propose the authors to extend the work adding an extra part where the possible utilization of this methodology can be applied - the related impact and benefits should be loudly presented.

Author Response

Dear Reviewer:

We gratefully appreciate your valuable suggestions. Our responses to the comments are shown below. The explanation part of the reply is in blue, and the revised manuscript content is in red. And revisions to the manuscript are marked up using the “Track Changes” function of MS Word version, such that any changes can be easily viewed by the editors and reviewers.

If there are any problems, please let us know. Thank you again.

Best regards,

Yours sincerely.

Prof. Yuping Wu(Corresponding author)

University of Chinese Academy of Sciences, Beijing 100049, China;

Institute of Microelectronics of Chinese Academy of Sciences, N0.03 Beitucheng West Road, Chaoyang District, Beijing, PR China. E-Mail: [email protected]

Phone: +86-010-82995693

Round 2

Reviewer 1 Report

My concern have been addressed, and I have no more comment.